# GABA uptake transporters support dopamine release in dorsal striatum with maladaptive downregulation in a parkinsonism model

Bradley M. Roberts [1,2✉], Natalie M. Doig [3], Katherine R. Brimblecombe [1,2], Emanuel F. Lopes [1],
Ruth E. Siddorn[1], Sarah Threlfell[1,2], Natalie Connor-Robson [1,2], Nora Bengoa-Vergniory [1,2],
Nicholas Pasternack[1], Richard Wade-Martins [1,2], Peter J. Magill [2,3] & Stephanie J. Cragg [1,2✉]

Striatal dopamine (DA) is critical for action and learning. Recent data show that DA release is under tonic inhibition by striatal GABA. Ambient striatal GABA tone on striatal projection neurons can be determined by plasma membrane GABA uptake transporters (GATs) located on astrocytes and neurons. However, whether striatal GATs and astrocytes determine DA output are unknown. We reveal that DA release in mouse dorsolateral striatum, but not nucleus accumbens core, is governed by GAT-1 and GAT-3. These GATs are partly localized to astrocytes, and are enriched in dorsolateral striatum compared to accumbens core. In a mouse model of early parkinsonism, GATs are downregulated, tonic GABAergic inhibition of DA release augmented, and nigrostriatal GABA co-release attenuated. These data define previously unappreciated and important roles for GATs and astrocytes in supporting DA release in striatum, and reveal a maladaptive plasticity in early parkinsonism that impairs DA output in vulnerable striatal regions.

[1] Centre for Integrative Neuroscience, Department of Physiology, Anatomy and Genetics, University of Oxford, Oxford OX1 3PT, UK. [2] Oxford Parkinson's Disease Centre, University of Oxford, Oxford OX1 3PT, UK. [3] Medical Research Council Brain Network Dynamics Unit, Nuffield Department of Clinical Neurosciences, University of Oxford, Oxford OX1 3TH, UK. ✉email: bradley.roberts@dpag.ox.ac.uk; stephanie.cragg@dpag.ox.ac.uk

Dopamine (DA) release in the dorsal and ventral striatum plays key roles in action selection and motivation, and is dysregulated in diverse disorders including Parkinson's disease (PD) and addictions. Striatal DA release is gated locally by axonal mechanisms and striatal neuromodulators that regulate or even drive DA release[1,2]. It has recently been revealed that DA release is under tonic inhibition by striatal GABA, operating through $GABA_A$ and $GABA_B$ receptors[3–5]. The striatum contains a high density of GABAergic projection neurons and interneurons and also receives a source of GABA co-released from mesostriatal DA neurons[6–8]. Given the paucity of GABAergic synapses on DA axons[9], tonic inhibition of DA release by striatal GABA is presumably mediated through extrasynaptic effects of ambient GABA[3] on receptors located presumably on DA axons. GABA can spillover for extrasynaptic actions on other nuclei[10], and in the dorsal striatum, provides a sizeable ambient GABA tone on spiny projection neurons (SPNs), evident as a tonic $GABA_A$ receptor-mediated inhibitory conductance[11–15].

Tonic inhibition by ambient GABA across the mammalian brain is usually limited by GABA uptake by plasma membrane GABA transporters (GATs)[16]. There are two isoforms of the GAT in striatum: GAT-1 (Slc6a1), abundant in axons of GABAergic neurons[17–20]; and GAT-3 (Slc6a11), expressed moderately[19–21] but observed particularly on astrocytes[19,22,23]. Emerging transcriptomic data additionally indicate that striatal astrocytes express both GAT-3 and GAT-1[22,24,25]. In addition, mRNA for GAT-1 and for GAT-3 has been found in midbrain DA neurons, and GATs have been suggested to be located on striatal DA axons to support GABA co-storage for co-release[7], with one corresponding report showing weak immunolabelling of GAT-1 on DA transporter (DAT)-positive structures in striatum[26]. Ambient GABA tone on SPNs in dorsal striatum is limited by GAT-1 and GAT-3[12–14,27], and recent evidence indicates that dysregulation of GAT-3 on striatal astrocytes results in profound changes to SPN activity and striatal-dependent behavior[23]. However, whether striatal GAT function and, by association, astrocytes are critical for setting the level of DA output has not previously been examined.

Here we reveal firstly that GAT-1 and GAT-3 strongly regulate striatal DA release in the dorsolateral striatum (DLS) but not in the nucleus accumbens core (NAcC), by limiting tonic inhibition arising from striatal ambient GABA. We also identify a role for GATs located on striatal astrocytes in supporting DA release, and furthermore, we reveal maladaptive reductions in GAT levels that impair DA output in the DLS in a mouse model of early parkinsonism.

## Results

**DA release in DLS and NAcC is tonically inhibited by a GAD-dependent GABA source.** We recently reported that axonal DA release in the dorsal striatum is under tonic inhibition by striatal GABA, as $GABA_A$ and $GABA_B$ receptor antagonists enhanced DA release evoked by single electrical and targeted optogenetic stimuli[3]. Since mechanisms that regulate striatal DA release can diverge between dorsal and ventral striatal territories[28–33], we first determined whether DA release in NAcC, within the ventral striatum, is similarly regulated by tonic GABA inhibition. We used fast-scan cyclic voltammetry (FSCV) in acute coronal slices of mouse brain to detect extracellular concentration of DA ($[DA]_o$) at carbon-fiber microelectrodes evoked optogenetically to activate DA axons selectively (Fig. 1a, b). Co-application of $GABA_A$ and $GABA_B$ receptor antagonists (+)-bicuculline (10 μM) and CGP 55845 (4 μM), respectively, significantly enhanced $[DA]_o$ evoked by single light pulses by ~25% in both DLS and NAcC, when compared to time-matched drug-free controls

(Fig. 1c, d; DLS: $F_{(1,16)} = 33.92$, $p < 0.0001$; two-way repeated-measures ANOVA; time-matched controls: $n = 8$ experiments/5 mice, $GABA_R$ antagonists: $n = 10$ experiments/6 mice; NAcC: $F_{(1,12)} = 20.68$, $p = 0.0007$; two-way repeated-measures ANOVA; time-matched controls: $n = 6$ experiments/5 mice, $GABA_R$ antagonists: $n = 8$ experiments/5 mice). These effects were similar in DLS and NAcC (Fig. 1e; $t_{(16)} = 1.089$, $p = 0.292$; unpaired Student's t-test) and were also observed when $[DA]_o$ was evoked by single electrical pulses (Supplementary Fig. 1a–c). This tonic GABAergic inhibition of DA release did not require cholinergic interneuron input to nAChRs (Supplementary Fig. 1a), or striatal glutamatergic input (Supplementary Fig. 1d), and was also seen when higher near-physiological bath temperatures of 37 °C were used (Supplementary Fig. 1e). These results confirm that DA release is under tonic inhibition by GABA in both ventral and dorsal striatal regions.

We tested whether GABAergic inhibition of DA release arose from GABA co-released by DA axons or from GABA originating from a canonical neuron source (i.e. striatal GABAergic neurons). Mesostriatal DA neurons synthesize, co-store and co-release GABA[6], with GABA synthesis depending on aldehyde dehydrogenase (ALDH)-1a1[8]. In contrast, canonical synthesis of GABA in neurons requires glutamic acid decarboxylase (GAD). We examined which source(s) of GABA is responsible for tonic inhibition of DA release using inhibitors of either ALDH or GAD. Pretreating slices with a nonselective ALDH inhibitor disulfiram (10 μM, 2–4 h) halved light-evoked $GABA_A$-dependent currents from DA axons onto SPNs (Supplementary Fig. 2a), as reported previously[8], but did not prevent GABA-receptor antagonists from enhancing DA release: in the DLS, GABA-receptor antagonists enhanced light-evoked $[DA]_o$ by ~40% in the presence of disulfiram, which was a significantly larger effect than seen without disulfiram (Fig. 1f; $F_{(12,168)} = 1.97$, $p = 0.030$; two-way repeated-measures ANOVA, drug × time interaction; Fig. 1h; $t_{(14)} = 2.923$, $p = 0.011$; unpaired Student's t-test; disulfiram: $n = 6$ experiments/5 mice, disulfiram absent: $n = 10$ experiments/6 mice). These data suggest that GABA co-released from DA axons is not responsible for tonic inhibition of DA release, and rather, that an ALDH-dependent source of GABA might act indirectly to limit tonic inhibition of DA by a different, ALDH-independent source. Disulfiram alone did not significantly modify evoked $[DA]_o$ (Supplementary Fig. 2b), but we caution that the effects of ALDH inhibition on DA release levels alone cannot be used to assess GABA tone because ALDH is also involved in DA catabolism. By contrast, when we pretreated brain slices with the GAD inhibitor 3-mercaptopropionic acid (3-MPA, 500 μM), which attenuates electrically evoked GABA transmission onto SPNs by more than half[8], the disinhibition of DA release in the DLS by GABA-receptor antagonists was attenuated (Fig. 1g; $F_{(1,16)} = 14.81$, $p = 0.0014$; two-way repeated-measures ANOVA; Fig. 1h; $t_{(16)} = 4.056$, $p = 0.0009$; unpaired Student's t-test; 3-MPA: $n = 8$ experiments/5 mice, 3-MPA absent: $n = 10$ experiments/6 mice), indicating that a GAD-dependent GABA source provides tonic inhibition of striatal DA release.

**GAT-1 and GAT-3 inhibition attenuates DA release in the DLS but not NAcC.** We tested the hypothesis that GATs, by governing ambient GABA[12–14,27], might determine the level of tonic inhibition of DA release. The nonselective GAT inhibitor (±)-nipecotic acid (NPA) (1–10 mM) inhibits all subtypes of GATs[34,35]. Bath application of NPA (1.5 mM) attenuated $[DA]_o$ by single electrical pulses in the DLS to ~60% of time-matched controls (Fig. 2a; $F_{(1,16)} = 73.40$, $p < 0.0001$; two-way repeated-measures ANOVA; NPA: $n = 9$ experiments/5 mice, time-matched controls: $n = 9$ experiments/7 mice). NPA also

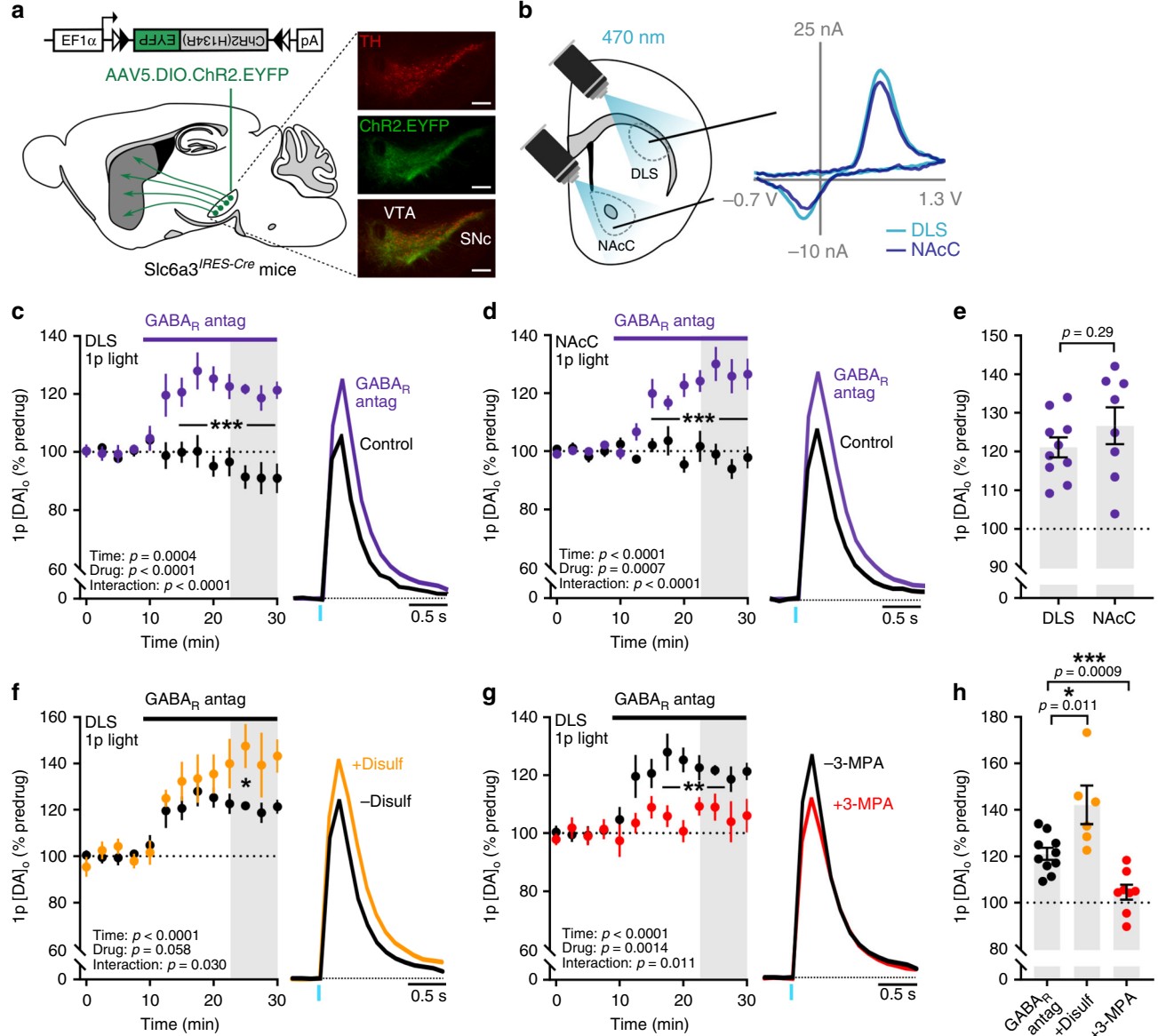

**Fig. 1 Striatal DA release is tonically inhibited by a GAD-dependent GABA source. a, b** Schematics representing the experimental configuration and representative voltammograms for light-evoked [DA]$_o$ in DLS and NAcC, after viral injection and expression of ChR2-eYFP in VTA and SNc in Slc6a3$^{IRES-Cre}$ mouse. TH (red), ChR2-eYFP (green). Scale bars: 0.25 mm. **c, d** Left, mean peak [DA]$_o$ during consecutive recordings evoked by a single light pulse (1p) in control conditions (black, $n = 8$ experiments/6 mice for DLS, $n = 6$ experiments/5 mice for NAcC) and with GABA$_A$ and GABA$_B$ receptor antagonists (solid bar)(purple, GABA$_R$ antag), (+)-bicuculline (10 μM) and CGP 55845 (4 μM), respectively, recorded in the DLS (**c**, $n = 10$ experiments/6 mice) or NAcC (**d**, $n = 5$ experiments/4 mice). Right, mean transients of [DA]$_o$ (normalized to predrug baselines) from last four timepoints (grey shaded region). **e** Mean peak [DA]$_o$ evoked by 1p light following GABA$_R$ antagonism in DLS and NAcC (as % of predrug baseline, data summarized from **c** and **d**). **f, g** Mean peak [DA]$_o$ during consecutive recordings evoked by 1p during application of GABA$_R$ antagonists in the absence (black, $n = 10$ experiments/6 mice) or the presence of ALDH inhibitor disulfiram (10 μM) (**f**, orange, $n = 6$ experiments/5 mice) or GAD inhibitor 3-MPA (500 μM) (**g**, red, $n = 7$ experiments/5 mice). **h** Mean peak [DA]$_o$ in DLS following GABA$_R$ antagonism in the absence or the presence of 3-MPA and disulfiram (as a % of predrug baseline, data summarized from **f** and **g**). Data are normalized to mean of four timepoints prior to GABA antagonist application (dotted line). Mean transients of [DA]$_o$ are derived from last four timepoints (grey shaded region) and normalized to predrug baselines. Two-way repeated-measures ANOVA with Sidak's multiple comparison tests (**c, d, f, g**) and two-tailed Student's unpaired $t$-tests (**e, h**). *$p < 0.05$, **$p < 0.01$, ***$p < 0.001$. Control data in **f** and **g** are the GABA$_R$ antagonist data from **c**. Error bars are ±SEM. Source data are provided as a Source Data file.

significantly reduced [DA]$_o$ evoked optogenetically by single light pulses (Supplementary Fig. 3a), indicating that attenuation of DA release does not require concurrent activation of other striatal neurons. NPA attenuated electrically evoked [DA]$_o$ to a greater extent in DLS than in NAcC (Fig. 2a–c; $t_{(13)} = 5.266$, $p = 0.0002$; unpaired Student's $t$-test) where NPA only marginally attenuated DA release compared to time-matched controls (Fig. 2b; $F_{(1,10)} =$

6.72, $p = 0.027$; two-way repeated-measures ANOVA; NPA: $n = 6$ experiments/4 mice, time-matched controls: $n = 6$ experiments/5 mice). These data indicate that the level of tonic inhibition of DA release is limited by GATs in DLS to a greater degree than in NAcC.

Two main isoforms of GATs are expressed in the basal ganglia: GAT-1 and GAT-3[36]. We used selective inhibitors of GAT-1 and

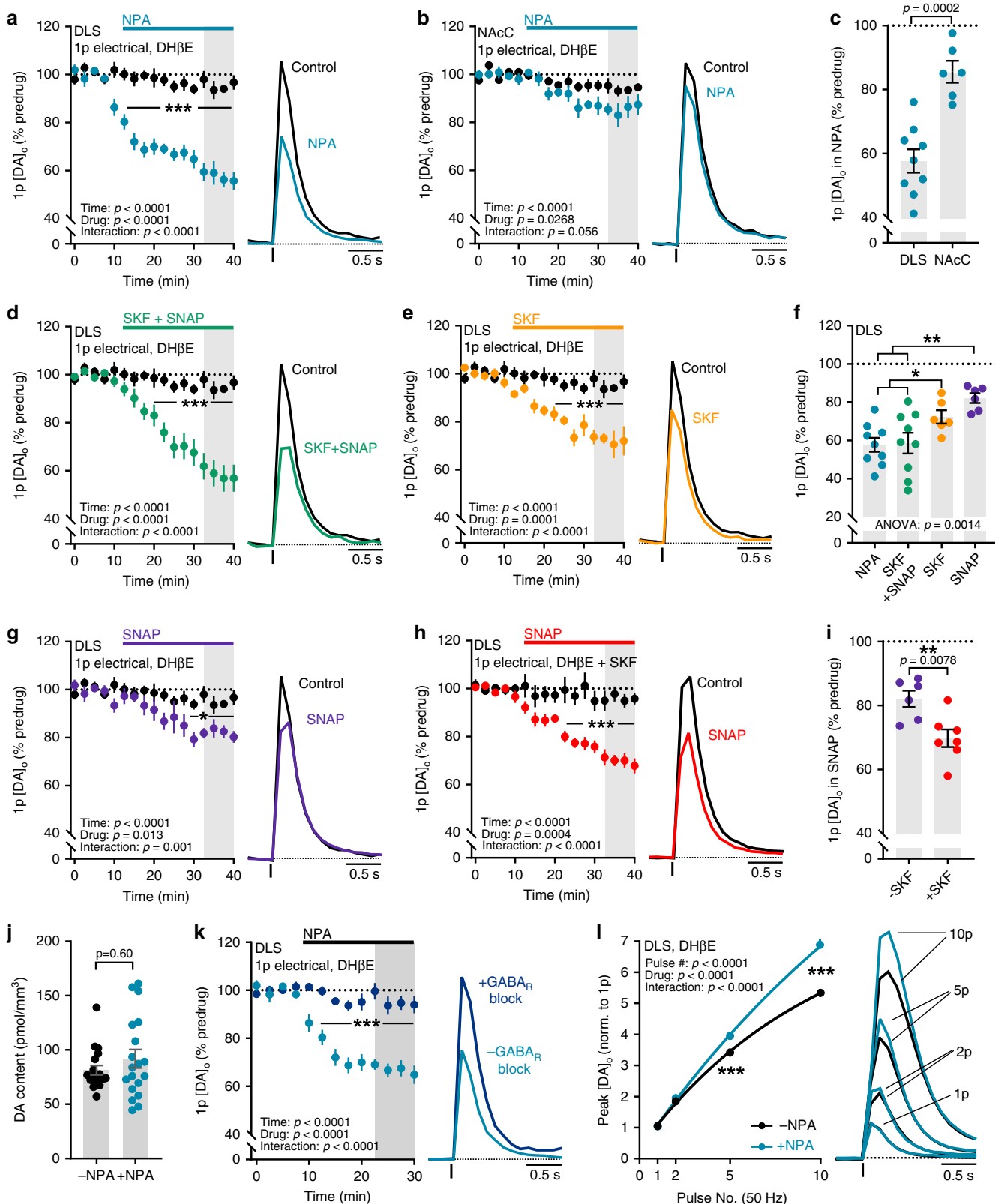

GAT-3 to identify which isoform(s) limit GABAergic inhibition of DA release in the DLS. Together, the combined inhibition of GAT-1 and GAT-3, with selective inhibitors SKF89976A (20 μM) and SNAP5114 (50 μM), respectively, significantly attenuated electrically evoked $[DA]_o$ to ~60% of time-matched controls

(Fig. 2d; $F_{(1,16)} = 24.79$, $p < 0.0001$; two-way repeated-measures ANOVA; SKF + SNAP: $n = 9$ experiments/5 mice), equivalent to that seen with broad-spectrum GAT inhibitor NPA (Fig. 2f; $F_{(3,26)} = 6.912$, $p = 0.0014$, one-way ANOVA; SKF + SNAP vs. NPA, $p = 0.9984$, Sidak's multiple comparisons). GAT-1

**Fig. 2 GAT-1 and GAT-3 inhibition attenuates DA release in DLS, but not NAcC. a, b, d, e, g, h** Mean peak $[DA]_o$ during consecutive recordings evoked by a single electrical pulse (*1p*) in DLS (**a, d, e, g, h**) or NAcC (**b**) in control conditions (black, $n = 9$ experiments/7 mice for DLS, $n = 6$ experiments/5 mice for NAcC) or with GAT inhibitor nipecotic acid (NPA, 1.5 mM) (**a**, *blue*, $n = 9$ experiments/5 mice; **b**, $n = 6$ experiments/4 mice), combined bath application of the GAT-1 specific inhibitor SKF89976A (20 μM) and the GAT-3 specific inhibitor SNAP5114 (50 μM) (**d**, *green*, $n = 9$ experiments/5 mice), SKF89976A alone (**e**, *orange*, $n = 6$ experiments/4 mice), SNAP5114 alone (**g**, *purple*, $n = 6$ experiments/4 mice), or bath application of SNAP5114 in slices preincubated in SKF89976A (**h**, *red*, $n = 7$ experiments/5 mice; black, SKF89976A preincubated controls, $n = 6$ experiments/3 mice). **c, f, i** Mean peak $[DA]_o$ evoked by 1p following GAT inhibition (expressed as a % of predrug baseline). **j** Mean DA content of dorsal striatum incubated in vehicle-treated control conditions (black, $n = 19$ punches/5 mice) or NPA (1.5 mM) (blue, $n = 19$ punches/5 mice). **k** Mean peak $[DA]_o$ during consecutive recordings evoked by 1p in DLS during application of NPA (1.5 mM) in the absence (light blue, $n = 9$ experiments/5 mice) or presence (dark blue, $n = 5$ experiments/4 mice) of $GABA_A$ (picrotoxin, 100 μM) and $GABA_B$ (CGP 55845, 4 μM) receptor antagonists. **l** Left, Mean peak values of $[DA]_o$ evoked by 50 Hz electrical pulses in DLS normalized to 1p in the absence (black, control, $n = 8$ experiments/5 mice) or presence of NPA (1.5 mM) (blue, $n = 8$ experiments/5 mice). Sigmoidal curve fits ($R^2 = 0.98$). Data are normalized to mean of four timepoints prior to GAT inhibitor application (dotted line); mean transients of $[DA]_o$ are derived from last four timepoints (grey shaded region) and normalized to predrug baselines. DHβE (1 μM) present throughout. Two-way repeated-measures ANOVA with Sidak's multiple comparison tests (**a, b, d, e, g, h, k, l**), two-tailed Student's unpaired *t*-tests (**c, i**), Mann–Whitney test (**j**) and one-way ANOVA with Sidak's multiple comparison tests (**f**). \*$p < 0.05$, \*\*$p < 0.01$, \*\*\*$p < 0.001$. Control data in **d, e, g** are the same as in **a**. Error bars are ±SEM. Source data are provided as a Source Data file.

inhibition alone with SKF89976A (20 μM) significantly attenuated evoked $[DA]_o$ in DLS to ~75% of time-matched controls (Fig. 2e; $F_{(1,13)} = 28.37$, $p = 0.0001$; two-way repeated-measures ANOVA; SKF: $n = 6$ experiments/4 mice) and GAT-3 inhibition alone with SNAP5114 (50 μM) significantly attenuated evoked $[DA]_o$ in DLS to ~80% of time-matched controls (Fig. 2g; $F_{(1,13)} = 8.205$, $p = 0.0133$; two-way repeated-measures ANOVA; SNAP: $n = 6$ experiments/4 mice), which were smaller effects compared to combined GAT-1 and GAT-3 inhibition (Fig. 2f; $F_{(3,26)} = 6.912$, $p = 0.0014$, one-way ANOVA; SKF vs. NPA, $p = 0.0192$; SKF vs. SKF + SNAP, $p = 0.0487$; SNAP vs. NPA, $p = 0.0031$; SNAP vs. SKF + SNAP, $p = 0.0045$; Sidak's multiple comparisons). The functional effects of GAT-3 inhibition on $GABA_A$ receptor-mediated tonic currents in SPNs have been shown to be compensated for by GAT-1-mediated GABA uptake, with GAT-3 function revealed better during GAT-1 inhibition[13,36]. We pretreated slices with GAT-1 inhibitor SKF89976A (20 μM) and revealed that subsequent bath application of GAT-3 inhibitor SNAP5114 (50 μM) attenuated electrically evoked $[DA]_o$ in DLS to ~70% of time-matched controls (Fig. 2h; $F_{(1,11)} = 25.09$, $p = 0.0004$; two-way repeated-measures ANOVA; SNAP with SKF pretreatment: $n = 7$ experiments/5 mice, corresponding SKF preincubated controls: $n = 6$ experiments/3 mice), a larger effect than with GAT-3 inhibitor alone (Fig. 2i; $t_{(11)} = 3.243$, $p = 0.0078$; unpaired Student's *t*-test). Altogether, these data show roles for both GAT-1 and GAT-3 in limiting the level of GABA inhibition of DA release in the DLS.

**GAT inhibition attenuates striatal DA release by increasing GABA-receptor tone.** We ruled out diminished DA storage as a cause of the attenuation of DA release following GAT inhibition: Striatal DA content measured using high-performance liquid chromatography (HPLC) with electrochemical detection was unchanged by incubation with GAT inhibitor NPA (Fig. 2j; $U = 162$, $p = 0.603$, Mann–Whitney test, $n = 19$ experiments/5 mice per condition). Instead, we confirmed that GAT inhibition modified DA release in a GABA-receptor-dependent manner. The acute effects of NPA on evoked $[DA]_o$ were prevented in the presence of antagonists for $GABA_A$ (picrotoxin, 100 μM) and $GABA_B$ (CGP 55845, 4 μM) receptors (Fig. 2k; $F_{(1,12)} = 40.41$, $p < 0.0001$; two-way repeated-measures ANOVA; without GABA-receptor antagonists: ~65% of baseline, $n = 9$ experiments/7 mice; with GABA-receptor antagonists: ~95% of baseline, $n = 5$ experiments/4 mice), consistent with GAT regulation of DA release being mediated via extracellular GABA acting on GABA receptors. In addition, we excluded roles in the effects of NPA of $D_2$ dopamine receptors, glutamate receptors, or modulation of

DA uptake, since NPA effects were preserved in the presence of respective inhibitors of each of these potential mechanisms (Supplementary Fig. 3b, c). We have previously shown that activation of striatal GABA receptors can slightly promote the activity-dependence of DA release during short stimulus trains[3]. Consistent with an increase in GABA-receptor activation, GAT inhibitor NPA increased the dependence of $[DA]_o$ on pulse number during 50 Hz pulse trains in DLS (Fig. 2l; $F_{(1,7)} = 128.9$, $p < 0.0001$; two-way repeated-measures ANOVA; $n = 8$ experiments/5 mice). NPA also increased the paired-pulse ratio of electrically evoked $[DA]_o$ at short inter-pulse intervals (Supplementary Fig. 3d, e) consistent with a decrease in DA release probability[37]. Together these data indicate that GAT inhibition attenuates DA release through increasing GABA-receptor tone and reducing DA release probability.

We assessed whether the greater role for GATs in regulating $[DA]_o$ in DLS than NAcC (see Fig. 2c) was due to differences in GABA-receptor regulation of DA. However, bath application of exogenous GABA (2 mM) attenuated $[DA]_o$ evoked by 1p electrical stimulation to a similar degree in DLS and NAcC (Supplementary Fig. 3f), arguing against a difference in GABA-receptor function as a major factor. These findings therefore suggest a different level of GAT function in limiting ambient GABA in DLS versus NAcC.

**GAT-1 and GAT-3 function and expression is enriched in DLS versus NAcC.** To identify whether GATs play a greater role in governing GABA tone in DLS than NAcC, we recorded the tonic $GABA_A$ receptor-mediated currents in SPNs using whole-cell voltage-clamp electrophysiology and assessed the impact of GAT inhibition on holding current. We confirmed that changes in holding current were mediated by $GABA_A$ receptors by subsequently applying $GABA_A$ receptor antagonist picrotoxin (PTX; 100 μM). Consistent with the differential effects on DA release, GAT inhibition with NPA (1.5 mM) promoted the $GABA_A$-mediated holding current in SPNs to a greater degree in DLS than in NAcC (Fig. 3a–c; DLS: $p = 0.0003$, Friedman's ANOVA on Ranks, NPA vs. drug-free baseline: $p = 0.001$, NPA + PTX vs. drug-free baseline: $p = 0.16$, NPA vs. NPA + PTX: $p < 0.001$, Student–Newman–Keuls tests, $n = 7$ cells/5 mice; NAcC: $p = 0.0001$, Friedman's ANOVA on Ranks, NPA vs. drug-free baseline: $p = 0.014$, NPA + PTX vs. drug-free baseline: $p = 0.014$, NPA vs. NPA + PTX: $p = 0.002$, Student–Newman–Keuls tests, $n = 6$ cells/3 mice; DLS vs NAcC: $U = 4$, $p = 0.0140$, Mann–Whitney test). These data corroborate a greater role for GATs in limiting ambient GABA tone in DLS than in NAcC.

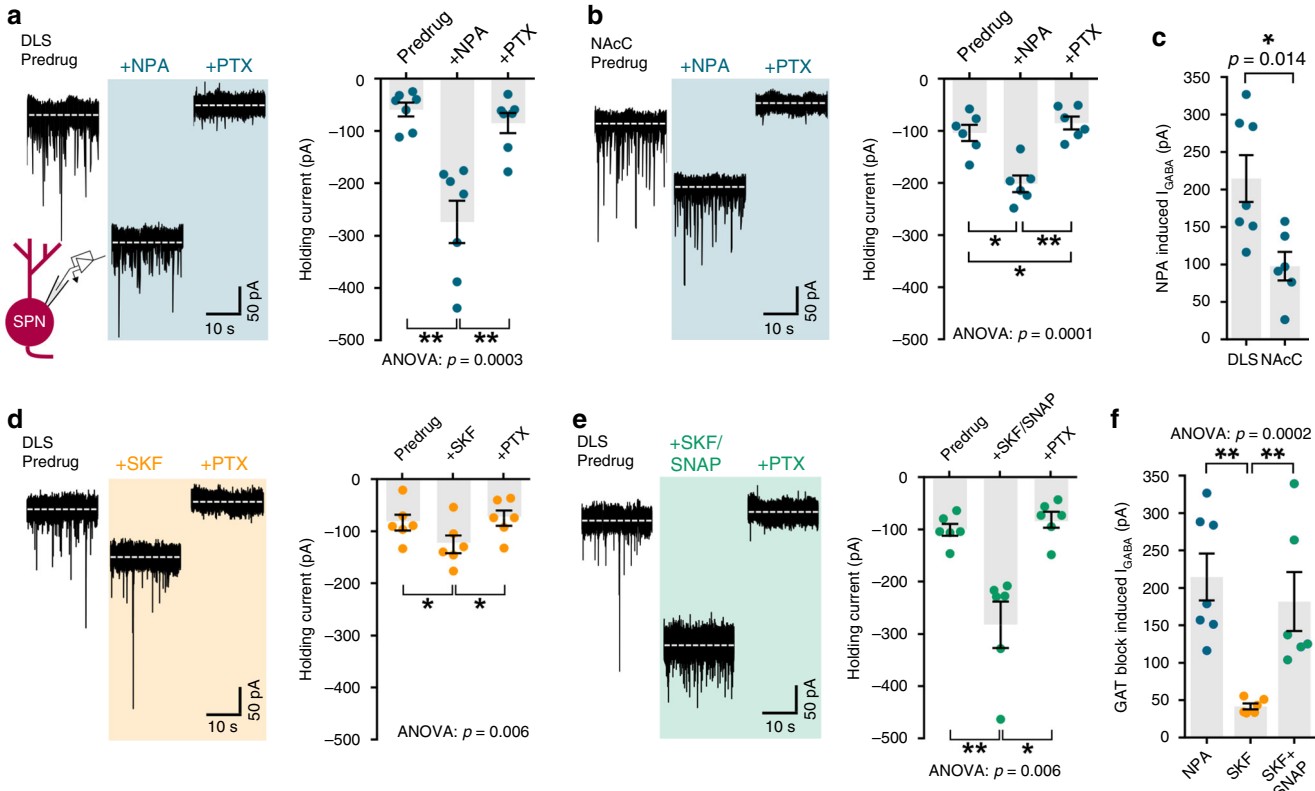

**Fig. 3 Tonic GABA currents in striatal spiny projection neurons (SPNs) are augmented by GAT inhibition. a, b, d, e** Left, representative continuous whole-cell recordings from SPNs in DLS (**a, d, e**) or NAcC (**b**) voltage clamped at −70 mV in the presence of ionotropic glutamate receptor antagonists NBQX (5 µM) and D-AP5 (50 µM), before and during bath application of GAT inhibitor NPA (blue, 1.5 mM, $n = 7$ cells/5 mice for DLS in **a**, $n = 6$ cells/3 mice for NAcC in **b**), GAT-1-specific inhibitor SKF89976A (orange, 20 µM, $n = 6$ cells/3 mice in **d**), or the combined application of SKF89976A and GAT-3-specific inhibitor SNAP5114 (green, 50 µM, $n = 6$ cells/4 mice in **e**). GAT inhibitors increase the extracellular $GABA_A$-mediated inward current, revealed by a shift in the holding current, and is reversed upon application of $GABA_A$ receptor antagonist picrotoxin (PTX, 100 µM). Right, mean holding current in pA recorded in SPNs in control conditions, upon addition of GAT inhibitors and then PTX. **c, f** Mean tonic $GABA_A$-receptor-mediated currents induced by GAT inhibition recorded from SPNs, calculated by subtracting predrug holding current from GAT block-induced holding current. Friedman's ANOVA on Ranks and Student–Newman–Keuls multiple comparisons (**a, b, d, e**), Mann–Whitney U test (**c**), Kruskal–Wallis test and Dunn's multiple comparisons (**f**). *$p < 0.05$, **$p < 0.01$. Error bars are ±SEM. Source data are provided as a Source Data file.

We found also that tonic GABA inhibition of SPNs in DLS, like DA release, was regulated by both GAT-1 and GAT-3. Inhibition of GAT-1 alone with SKF89976A (20 µM) induced a small increase in the $GABA_A$-mediated holding current (Fig. 3d; $p = 0.006$, Friedman's ANOVA on Ranks, SKF vs. drug-free baseline: $p = 0.001$, SKF + PTX vs. drug-free baseline: $p = 0.41$, SKF vs. SKF + PTX: $p = 0.01$, Student–Newman–Keuls tests, $n = 6$ cells/3 mice). Combined inhibition of GAT-1 and GAT-3 with SKF89976A (20 µM) and SNAP5114 (50 µM) induced a three-fold increase (Fig. 3e; $p = 0.006$, Friedman's ANOVA on Ranks, SKF + SNAP vs. drug-free baseline: $p = 0.001$, SKF + SNAP + PTX vs. drug-free baseline: $p = 0.41$, SKF + SNAP vs. SKF + SNAP + PTX: $p = 0.011$, Student–Newman–Keuls tests, $n = 6$ cells/4 mice), which was greater than after GAT-1 inhibition alone, but similar to that seen with broad-spectrum GAT inhibition by NPA (Fig. 3f; $p = 0.0002$, Kruskal–Wallis ANOVA; SKF + SNAP vs. SKF: $p < 0.01$, NPA vs. SKF + SNAP: $p > 0.05$, NPA vs. SKF: $p < 0.01$; Dunn's multiple comparison tests). These effects of GAT inhibition were due to GATs limiting an action potential-independent GABA tone i.e. due to "spontaneous" GABA release[27] since in the presence of Na$_v$ blocker tetrodotoxin (TTX, 1 µM), NPA increased the $GABA_A$-mediated holding current in SPNs in the DLS to a similar level to that induced in TTX-free conditions (Supplementary Fig. 4).

Collectively, these results show that striatal GAT-1 and GAT-3 regulate an ambient GABA tone, and to a greater degree in DLS than in NAcC. We explored an anatomical basis for this regional heterogeneity in GAT function. Striatal immunoreactivity to GAT-1 and GAT-3 in the DLS and NAcC revealed a modest relative enrichment in the DLS for both GAT-1 (Fig. 4a, b; $p = 0.0093$, Wilcoxon signed-rank test, $n = 12$ hemispheres/6 mice) and GAT-3 (Fig. 4c, d; $p = 0.0015$, Wilcoxon signed-rank test, $n = 12$ hemispheres/6 mice). We also noted enriched GAT-3 in the medial NAc shell (NAcS) contiguous with the medial septal nucleus (Supplementary Fig. 5a–d). This observation prompted us to test the effects of GAT inhibition on DA release in NAcS. Correspondingly, GAT inhibition diminished electrically evoked [DA]$_o$ in NAcS to a greater degree than in NAcC (Supplementary Fig. 5e–g), indicating further regional heterogeneity in the role of GATs in limiting tonic inhibition. We note also that while GAT levels and function tally, they do not necessarily match the effect size of GABA-receptor antagonists in each striatal region, suggesting that additional mechanisms besides GATs regulate GABA tone and differ between regions.

**GAT-1 and GAT-3 on astrocytes are key regulators of ambient GABA inhibition of DA release.** Striatal GATs are located on the

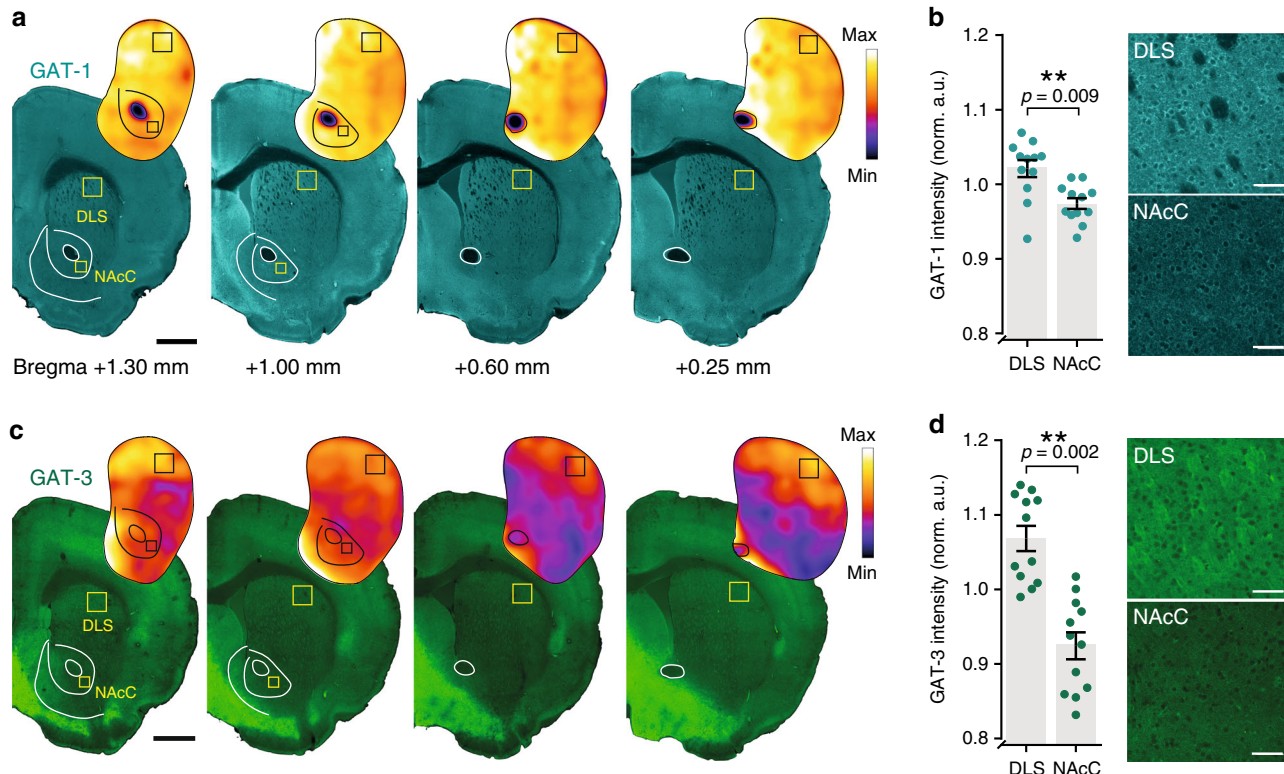

**Fig. 4 Enrichment of GAT-1 and GAT-3 expression in the DLS versus NAcC. a, c** Representative immunofluorescence signals for GAT-1 (cyan, **a**) and GAT-3 (green, **c**) using confocal microscopy in coronal sections across the rostral-caudal limits containing striatum prepared from an individual C57BL/6 J mouse with heatmaps for striatal GAT intensity. Boxes indicate representative locations for GAT intensity measurements in the dorsolateral striatum (DLS) and nucleus accumbens core (NAcC). Scale bars: 1 mm. Note enriched GAT-3 in the medial NAc shell contiguous with the medial septal nucleus and enriched GAT-3 expression in the claustrum. **b, d** Left, Mean GAT-1 (**b**) and GAT-3 (**d**) intensity in DLS and NAcC normalized to total striatum and averaged across rostral-caudal sites for each hemisphere ($n = 12$ hemispheres/6 mice for each GAT-1 and GAT-3). *Right*, Representative single plane images of GAT-1 (**b**) and GAT-3 (**d**) immunofluorescence from DLS and NAcC; imaging parameters were kept constant across regions. Scale bars: 50 μm. Mann–Whitney U tests (**b, d**). **$p < 0.01$. Error bars are ±SEM. Source data are provided as a Source Data file.

plasma membranes of cells that include GABAergic neurons[17–20] and astrocytes[19,22,23]. GAT-1 and GAT-3 have also been suggested to reside on DA axons to support GABA uptake for co-release[7]. To better understand where GATs are located to regulate tonic GABAergic inhibition of DA release, we probed two of these candidate locations, namely DA axons and astrocytes. We explored whether GAT-1 or GAT-3 could be detected on DA axons using immunofluorescence and confocal microscopy, but we did not find robust evidence to support their localization on DA axons conditionally expressing an eYFP reporter (Supplementary Fig. 6). As a positive control for GAT-1 detection, we confirmed that GAT-1 could however be localized to the neurites of parvalbumin (PV)-expressing GABAergic interneurons (Supplementary Fig. 7), which express GAT-1[17].

In many brain regions, including striatum, astrocytes are thought to regulate ambient GABA levels through uptake[23]. GAT-3 protein expression has been well documented on striatal astrocytes[19,22,23], and although GAT-1 is typically associated with neuronal structures[38], recent transcriptomic studies have found RNA for both GAT-1 and GAT-3 in striatal astrocytes[22,24,25]. We revisited GAT localization to astrocytes, using immunofluorescence and confocal microscopy with antibodies directed against either GAT-1 or GAT-3, as well as against the striatal astrocytic marker S100β[22] (Fig. 5a, b) in the DLS and NAcC. As expected, GAT-3 could be colocalized to S100β-expressing astrocytes (Fig. 5d), where GAT-3-immunoreactivity was observed distributed over a large surface area of plasma membrane when assessed in three dimensions (Supplementary Fig. 8d–f). We also found

several instances of similar localization of GAT-1-immunoreactivity on S100β-expressing astrocytes (Fig. 5c, Supplementary Fig. 8a–c). These data indicate that GAT-3 and GAT-1 proteins can be expressed on the plasma membranes of striatal astrocytes.

Given the presence of GAT-3 and GAT-1 on striatal astrocytes, we probed whether astrocytes participate in regulating the level of inhibition of DA release by GABA. We exposed striatal slices to the gliotoxin fluorocitrate, which inhibits the enzyme aconitase, in turn disrupting the tricarboxylic acid cycle and inducing metabolic arrest in astrocytes[39–41]. This approach has previously been established to render astrocytes inactive and prevent the effects of astrocytic GAT[42,43]. We pretreated slices with fluorocitrate (200 μM for 45–60 min) or vehicle and then co-incubated with or without NPA (1.5 mM for 30 min), and assessed effect on [DA]$_o$ evoked by 1p electrical stimulation across a range of sites in the DLS. We first confirmed that we could detect the effects of GAT inhibition in DLS in vehicle-treated control slices. Accordingly, [DA]$_o$ evoked from slices incubated in NPA was significantly less than those incubated in NPA-free control conditions, as expected (Fig. 6a; $U = 116$, $p = 0.00003$, Mann–Whitney test, $n = 24$ observations/5 mice for each condition), and the 4p/1p ratio (50 Hz) was appropriately enhanced (Fig. 6b; $t_{(14)} = 2.988$, $p = 0.009$; unpaired Student's t-test, $n = 8$ experiments/4 mice for each condition). By contrast, when we performed these experiments in slices pretreated with fluorocitrate to inactivate astrocytes, NPA did not significantly modify [DA]$_o$ evoked by 1p (Fig. 6c; $U = 699$, $p = 0.103$,

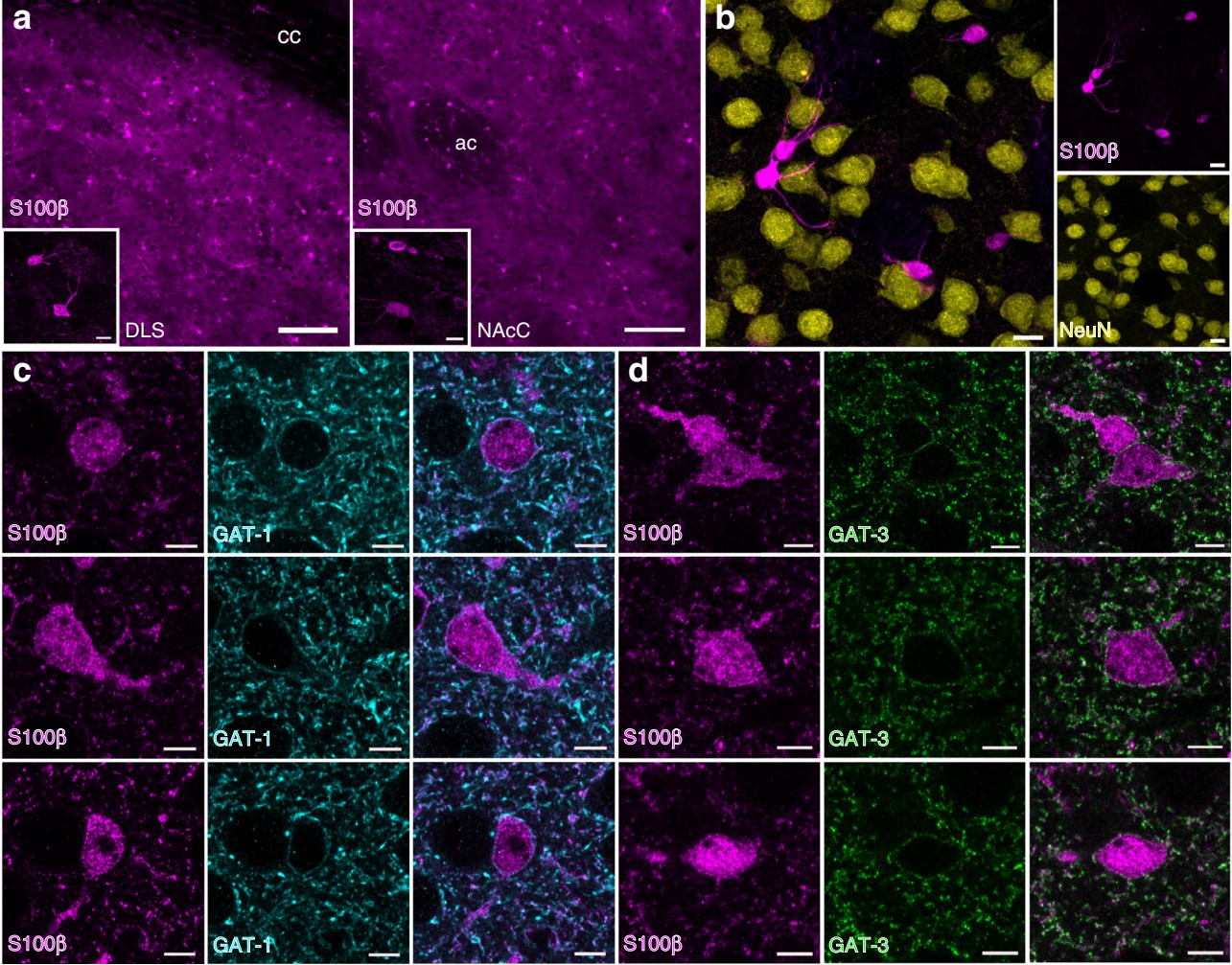

**Fig. 5 GAT-1 and GAT-3 are expressed on plasma membranes of striatal astrocytes. a** Striatal immunofluorescence signals for astrocyte marker S100β (magenta) in dorsolateral striatum (DLS) and nucleus accumbens core (NAcC). Scale bars: 100 μm, for inset: 10 μm. cc corpus callosum, ac anterior commissure. **b** Immunofluorescence signals for S100β do not colocalize with immunofluorescence signals for neuronal marker NeuN. Scale bars: 10 μm. **c**, **d** GAT-1 (cyan, **c**) and GAT-3 (green, **d**) and are expressed on plasma membranes of striatal S100β-expressing astrocytes imaged in DLS. Immunoreactivity for left, S100β, centre, GAT, right, merged. Scale bars: 5 μm.

Mann–Whitney test, $n = 42$ observations/7 mice for each condition), or the 4p/1p ratio (50 Hz), compared to NPA-free conditions (Fig. 6d; $t_{(24)} = 0.5384$, $p = 0.595$; unpaired Student's $t$-test, $n = 13$ experiments/7 mice for each condition). The effect of NPA on $[DA]_o$ when astrocytes were inhibited was significantly less than when astrocytes were intact (Fig. 6e; $U = 288$, $p = 0.0036$, Mann–Whitney test). We also noted that a comparison of evoked $[DA]_o$ with and without fluorocitrate revealed that $[DA]_o$ was reduced by fluorocitrate treatment (Fig. 6f; $U = 226$, $p = 0.0001$, Mann–Whitney test). Together, these data suggest that astrocytic GATs support GABA uptake and limit the level of inhibition of DA release by ambient GABA, such that in turn, astrocytes indirectly support DA release.

**Tonic inhibition of DA release in the DLS is augmented in a mouse model of parkinsonism.** Our data provide compelling evidence that GAT function regulates DA output level in the DLS. Dysregulation of GATs in the basal ganglia, including on astrocytes, has now been implicated in several models of neurological disease: in 6-OHDA toxin-based rat and mouse models of dopamine depletion in Parkinson's, astrocytes in the external globus pallidus have downregulated GAT-3[44]; and in R6/2 and FVB/N transgenic mouse models of Huntington's disease, striatal

GAT expression is increased and tonic inhibition by ambient GABA decreased[15,27], with one recent study directly implicating GATs on astrocytes in this mechanism[23]. Given that deficits in DA transmission in DLS, but not in NAcC, are common to transgenic rodent models of parkinsonism prior to cell loss[32,45,46], we explored whether tonic GABAergic inhibition of striatal DA release and its regulation by striatal GATs might be dysregulated in a mouse model of early parkinsonism.

We chose the *SNCA*-OVX mouse, a BAC-transgenic mouse model of early parkinsonism[32]. *SNCA*-OVX mice are devoid of mouse α-synuclein but overexpress human wild-type α-synuclein at disease-relevant levels and show early deficits in DA release prior to DA cell loss[32]. The littermate control mice for the *SNCA*-OVX model are devoid of mouse and human α-synuclein[32,47] and do not differ in DA release from wild-type mice using these protocols[48]. We made *SNCA*-OVX mice and their α-synuclein-null littermate controls optogenetics-capable by crossing with *Slc6a3*[IRES-Cre] α-synuclein-null mice, such that they allowed for optical manipulation of DA axons. Consequently, the two genotypes generated were: (1) "SNCA+" mice that express *Cre* recombinase in DA neurons and the human α-synuclein transgene and are mouse α-synuclein-null; and (2) their littermate controls, "Snca−/−" mice, that express *Cre*

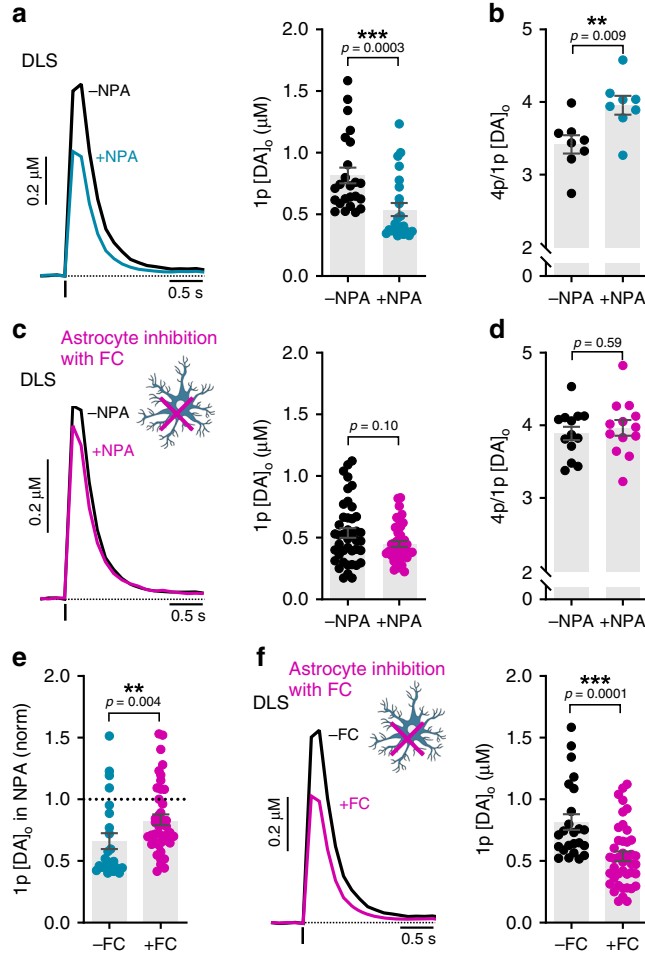

**Fig. 6 GATs on striatal astrocytes regulate GABA inhibition of DA release in DLS. a–d** Mean profiles of [DA]$_o$ and mean peak [DA]$_o$ in DLS evoked by a single electrical pulse (*1p*) (**a**, **c**) or four pulses normalized to 1p (**b**, **d**) in the absence (black) and presence of GAT inhibitor NPA (blue or pink, 1.5 mM) in vehicle-treated control slices (**a**, *n* = 24 observations/5 mice; **b**, *n* = 9 observations/5 mice) or in slices treated with astrocyte inhibitor fluorocitrate (FC, 200 μM) (**c**, *n* = 42 observations/7 mice; **d**, *n* = 13 observations/7 mice). **e** Mean peak [DA]$_o$ evoked by 1p in the presence of NPA (1.5 mM) normalized to control conditions in control slices (blue) or fluorocitrate-treated slices (pink) from **a** and **c**. **f** Mean peak [DA]$_o$ evoked by 1p in the absence of NPA in control slices (−*FC*) and in fluorocitrate-treated slices (+*FC*) from **a** and **b**. DHβE (1 μM) present throughout. Mann–Whitney U tests (**a, c, e, f**) and two-tailed Student's unpaired *t*-tests (**b, d**). ***p* < 0.01, ****p* < 0.001. Error bars are ±SEM. Source data are provided as a Source Data file.

recombinase in DA neurons and are mouse α-synuclein-null, and have no human α-synuclein transgene.

We first confirmed that, as observed in the original *SNCA*-OVX mice[32], the *SNCA+* mice at 4 months of age exhibited a ~30% deficit in electrically evoked [DA]$_o$ when compared to littermate controls (*Snca−/−*) in the dorsal striatum (Fig. 7a; $t_{(46)} = 3.272$, $p = 0.0020$; unpaired Student's *t*-test, *n* = 24 observations/5 mice for each genotype) but not in the NAc (Fig. 7a; $t_{(40)} = 1.393$, $p = 0.1714$; unpaired Student's *t*-test; *n* = 21 observations/5 mice for each genotype). The DA release deficit was also not attributable to any change in striatal DA content in *SNCA+* mice compared to *Snca−/−* mice (Fig. 7b; dorsal striatum: $t_{(14)} = 0.1625$, $p = 0.8733$, *n* = 8 experiments/5 mice

for each genotype; NAc: $t_{(14)} = 0.7445$, $p = 0.4689$, unpaired Student's *t*-tests; *n* = 8 experiments/5 mice for each genotype) establishing an underlying change to DA release ability rather than storage potential. We then verified that [DA]$_o$ evoked optogenetically in DLS by single light pulses showed a similar deficit in *SNCA+* compared to *Snca−/−* (Fig. 7c, d; $t_{(29)} = 2.443$, $p = 0.0209$, unpaired Student's *t*-test; $F_{(1,12)} = 7.108$, $p = 0.0206$; two-way repeated-measures ANOVA; *SNCA+*: *n* = 16 observations/3 mice; *Snca−/−*: *n* = 15 observations/3 mice). Having established a deficit in DA release in this optogenetics-capable mouse model of parkinsonism, we next addressed whether DA release deficits are accompanied by corresponding deficits in GABA co-release from DA axons. Using voltage-clamp recordings in SPNs, we observed a significantly lower amplitude of IPSCs evoked by light-activation of DA axons in *SNCA+* compared to *Snca−/−* mice (Fig. 7e, f; $t_{(14)} = 2.680$, $p = 0.0179$, unpaired Student's *t*-test; $F_{(1,14)} = 7.281$, $p = 0.0173$; two-way repeated-measures ANOVA; *SNCA+*: *n* = 7 cells/4 mice; *Snca−/−*: *n* = 9 cells/4 mice) indicating a companion deficit in GABA release, and these IPSCs exhibited a similar gradual rundown to DA release (Fig. 7d, f). Light-evoked IPSCs were GABA$_A$ receptor-mediated as they were eliminated by picrotoxin (PTX, 100 μM) (Supplementary Fig. 9a, b) and the observed differences in IPSC amplitudes were not due to differences in series resistance (Supplementary Fig. 9c).

We then explored in this model whether tonic GABA inhibition of DA release was modified in DLS or NAcC. We found that GABA$_R$ antagonism enhanced [DA]$_o$ evoked by single light pulses to a significantly greater degree in *SNCA+* mice than in *Snca−/−* controls in DLS (Fig. 8a; $F_{(1,13)} = 12.42$, $p = 0.0037$; two-way repeated-measures ANOVA; *SNCA+*: *n* = 8 experiments/5 mice; *Snca−/−*: *n* = 7 experiments/5 mice) but not in NAcC (Fig. 8b; $F_{(1,15)} = 2.318$, $p = 0.1487$; two-way repeated-measures ANOVA; *SNCA+*: *n* = 8 experiments/5 mice; *Snca−/−*: *n* = 9 experiments/ 5 mice), which was a significant regional difference (Fig. 8c, d; $p < 0.0001$, Kolmogorov–Smirnov test; U = 10, $p = 0.0207$, Mann–Whitney test). These data indicate that tonic GABA inhibition of DA axons is elevated in the DLS of *SNCA+* mice.

We tested the hypothesis that elevated tonic inhibition of DA release in the DLS of *SNCA+* mice might be due to impaired GAT function. We identified that there was an impairment in the effect of the nonselective GAT inhibitor NPA on DA release in DLS: there was an attenuated effect of NPA on [DA]$_o$ evoked by single electrical pulses in *SNCA+* versus *Snca−/−* controls (Fig. 8e–g; $F_{(1,8)} = 5.790$, $p = 0.0428$; two-way repeated-measures ANOVA; $p < 0.0001$, Kolmogorov–Smirnov test; $t_{(8)} = 3.244$, $p = 0.0118$, unpaired Student's *t*-test; *SNCA+*: *n* = 5 experiments/4 mice; *Snca−/−*: *n* = 5 experiments/4 mice). Furthermore, quantification of Western blots of dorsal striatal tissue revealed significantly lower levels of both GAT-1 and GAT-3 proteins in *SNCA+* mice versus *Snca−/−* controls (Fig. 8h; GAT-1: U = 2, $p = 0.0004$; GAT-3: U = 10, $p = 0.0136$, Mann–Whitney tests; *n* = 7 *SNCA+* mice, *n* = 10 *Snca−/−* mice). Given the expression and role of both GATs on astrocytes in regulating DA release, these data strongly implicate GAT downregulation on astrocytes as a contributing cause of enhanced GABA inhibition of DA release in *SNCA+* mice. We therefore tested whether by impairing astrocytic GAT function through astrocyte inactivation with fluorocitrate, we might equalise the level of tonic GABA inhibition operating in the DLS of *SNCA+* and *Snca−/−* mice. In slices preincubated with fluorocitrate (200 μM for 45–60 min), the difference in the effect of GABA$_R$ antagonists on [DA]$_o$ evoked by single light pulses seen in DLS between *SNCA+* mice and *Snca−/−* controls was completely abolished, and the effect of GABA$_R$-antagonists was not different between genotypes (Fig. 8i; $F_{(1,15)} = 0.005$, $p = 0.946$; two-way repeated-measures ANOVA;

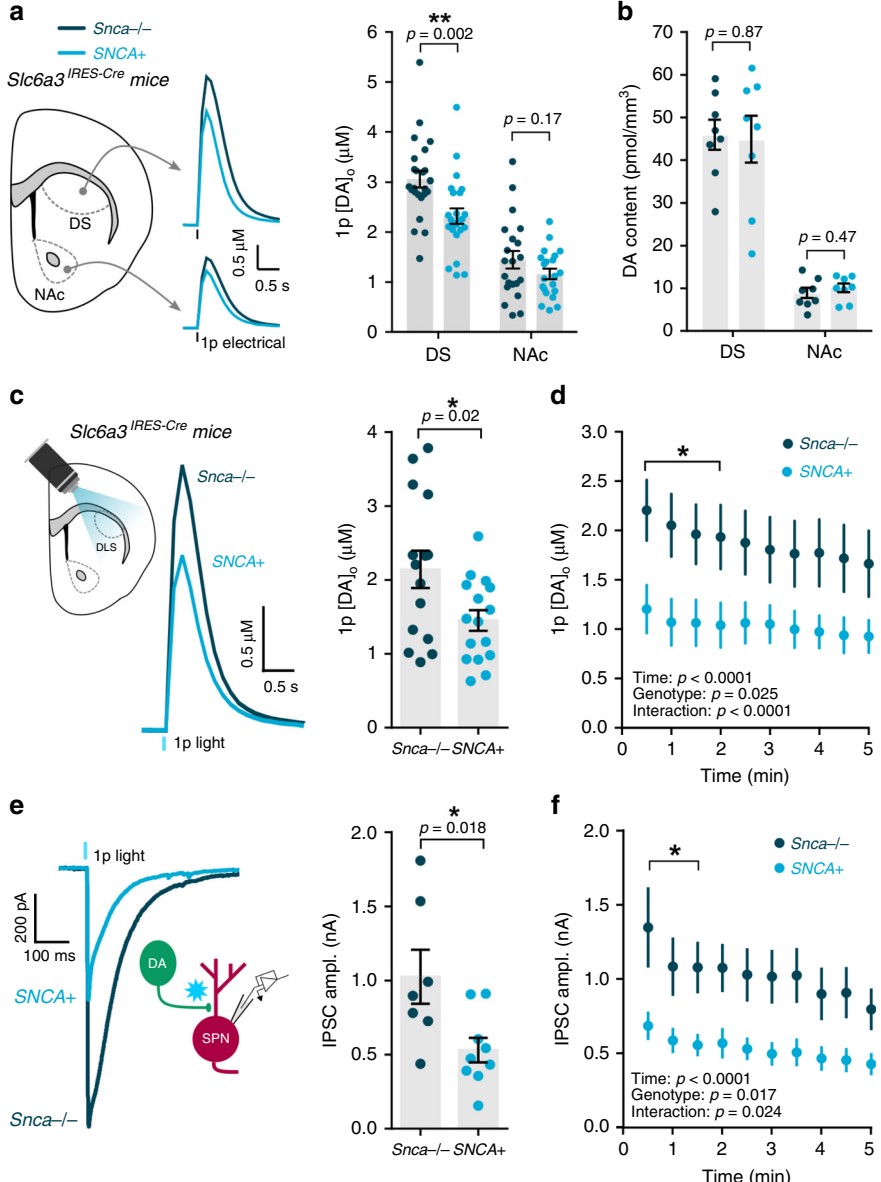

**Fig. 7 Attenuated striatal GABA and DA co-release from DA axons in mouse model of early parkinsonism. a** Left, mean [DA]$_o$ profiles vs. time evoked by a single pulse (*1p*) of electrical stimulation in dorsal striatum (DS) and nucleus accumbens (NAc) of *SNCA+* mice (light blue) and littermate controls (*Snca−/−*, dark blue) at 3–4 months, backcrossed onto an *Slc6a3*$^{IRES-Cre}$ background. Right, Mean 1p-evoked [DA]$_o$ (in μM) from *Left* (*n* = 24 observations/5 mice per genotype in DS, *n* = 21 observations/5 mice per genotype in NAc). **b** Mean DA content in DS and NAc of *SNCA+* mice (light blue) and littermate controls (*Snca−/−*, dark blue) (*n* = 8 experiments/5 mice per genotype in DS and NAc). **c** Left, mean [DA]$_o$ profiles vs. time following 1p light simulation in DLS of *SNCA+* mice (light blue) and littermate controls (*Snca−/−*, dark blue). Right, Mean 1p-evoked [DA]$_o$ (in μM) from Left (*n* = 15 observations/3 mice in *Snca−/−* mice, *n* = 16 observations/3 mice in *SNCA+* mice). **d** Mean 1p light-evoked [DA]$_o$ (in μM) recorded every 30 s in DLS of *SNCA+* mice (light blue, *n* = 7 experiments/3 mice) and littermate controls (*Snca−/−*, dark blue, *n* = 7 experiments/3 mice). **e, f** Mean 1p light-evoked inhibitory postsynaptic currents (IPSCs) recorded from spiny projection neurons (SPNs) every 30 s in the DLS of *SNCA+* mice (light blue, *n* = 9 cells/4 mice) and littermate controls (*Snca−/−*, dark blue, *n* = 7 cells/4 mice), voltage clamped at −70 mV and in the presence of ionotropic glutamate receptor antagonists (NBQX, 5 μM; D-APV, 50 μM). Two-tailed Student's unpaired *t*-tests (**a–c, e**) and two-way repeated-measures ANOVA with Sidak's multiple comparison tests (**d, f**). *$p$ < 0.05, **$p$ < 0.01. Error bars are ±SEM. Source data are provided as a Source Data file.

*SNCA+*: *n* = 8 experiments/3 mice; *Snca−/−*: *n* = 9 experiments/ 3 mice). Fluorocitrate incubation significantly boosted the effect of GABA$_R$ antagonists in *Snca−/−* mice (Fig. 8j, k; $p$ < 0.0001, Kolmogorov–Smirnov test; t$_{(14)}$ = 5.82, $p$ < 0.0001, unpaired Student's *t*-test) but had no effect in *SNCA+* mice (Fig. 8k; t$_{(14)}$ = 0.44, $p$ = 0.664, unpaired Student's *t*-test). Taken together, these data suggest that tonic inhibition of DA release by ambient GABA is augmented in the dorsal striatum in early parkinsonism

due to downregulation of GATs that are located at least in part to astrocytes (Fig. 9).

## Discussion

We define a major role for striatal GATs and astrocytes in setting the level of DA output in the striatum. We show that GAT-1 and GAT-3, located at least in part on striatal astrocytes, govern tonic GABAergic inhibition of DA release. GATs operate in a

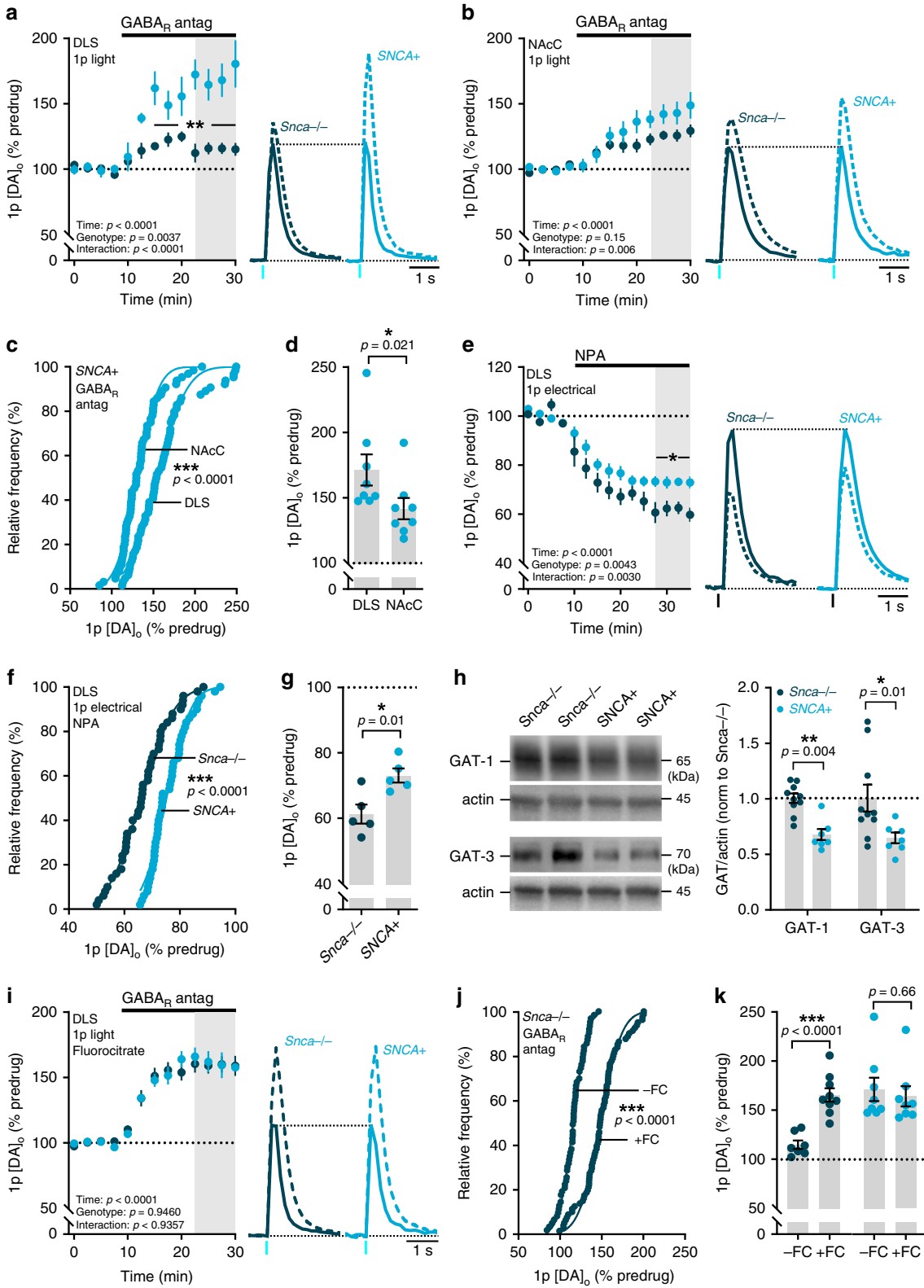

heterogeneous manner across the striatum, substantially limiting tonic inhibition of DA release in DLS but not NAcC. Moreover, in a mouse model of early parkinsonism, we reveal maladaptive decreases in striatal GAT-1 and GAT-3 expression and consequently, profound augmentation of tonic inhibition of DA release by GABA in the dorsal striatum.

We found that tonic inhibition of DA release by GABA spans dorsal-ventral territories of striatum, and arises from a GAD-dependent source of GABA. The source of GABA was not a non-canonical ALDH-dependent source e.g. co-release from DA axons, because inhibition of ALDH did not attenuate the tonic inhibition of DA release by GABA, despite attenuating GABA

**Fig. 8 Enhanced tonic inhibition of striatal DA release and impaired GAT function in mouse model of early parkinsonism. a, b, e, i** Mean peak $[DA]_o$ during consecutive recordings evoked by a single (1p) light (**a**, **b**, **i**) or electrical pulse (**e**) in DLS (**a**, **e**, **i**) or NAcC (**b**) during applications of antagonists for $GABA_A$ (bicuculline, 10 μM) and $GABA_B$ receptors (CGP 55845, 4 μM) (**a**, **b**, **i**), or the nonspecific GAT inhibitor NPA (1.5 mM) (**e**) in slices preincubated (**i**) or not preincubated (**a**, **b**, **e**) with fluorocitrate (FC, 200 μM, 45–60 min) from Snca−/− (dark blue, $GABA_R$ antagonism: n = 7 experiments/5 mice in DLS (**a**), n = 9 experiments/5 mice in NAcC (**b**), n = 7 experiments/3 mice for fluorocitrate (**i**); NPA: n = 5 experiments/4 mice (**e**) and SNCA+ mice (light blue, $GABA_R$ antagonism: n = 8 experiments/5 mice for both DLS and NAcC (**a**, **b**), n = 6 experiments/3 mice for fluorocitrate (**i**); NPA: n = 5 experiments/4 mice (**e**). Data are normalized to mean of four timepoints prior to drug application (dotted line); mean transients of $[DA]_o$ are derived from last four timepoints (grey shaded region) and normalized to predrug baselines. **c, d, f, g, j, k** Cumulative frequency plots of individual data points (**c**, **f**, **j**) and mean per recording site (**d**, **g**, **k**) from **a**, **b**, **e**, **i**. **h** Representative Western blots and mean GAT-1 and GAT-3 protein content of dorsal striatum tissue taken from Snca−/− mice (n = 10 mice) and SNCA+ mice (n = 7 mice). Data normalized to actin and littermate control expression. Two-way repeated-measures ANOVA with Sidak's multiple comparison tests (**a**, **b**, **e**, **i**), Komogorov–Smirnov tests (**c**, **f**, **j**), two-tailed Student's unpaired t-tests (**d**, **g**, **k**) and Mann–Whitney U tests (**h**). $*p < 0.05$, $**p < 0.01$, $***p < 0.001$. Error bars are ±SEM. Source data and uncropped blots from **h** are provided as a Source Data file.

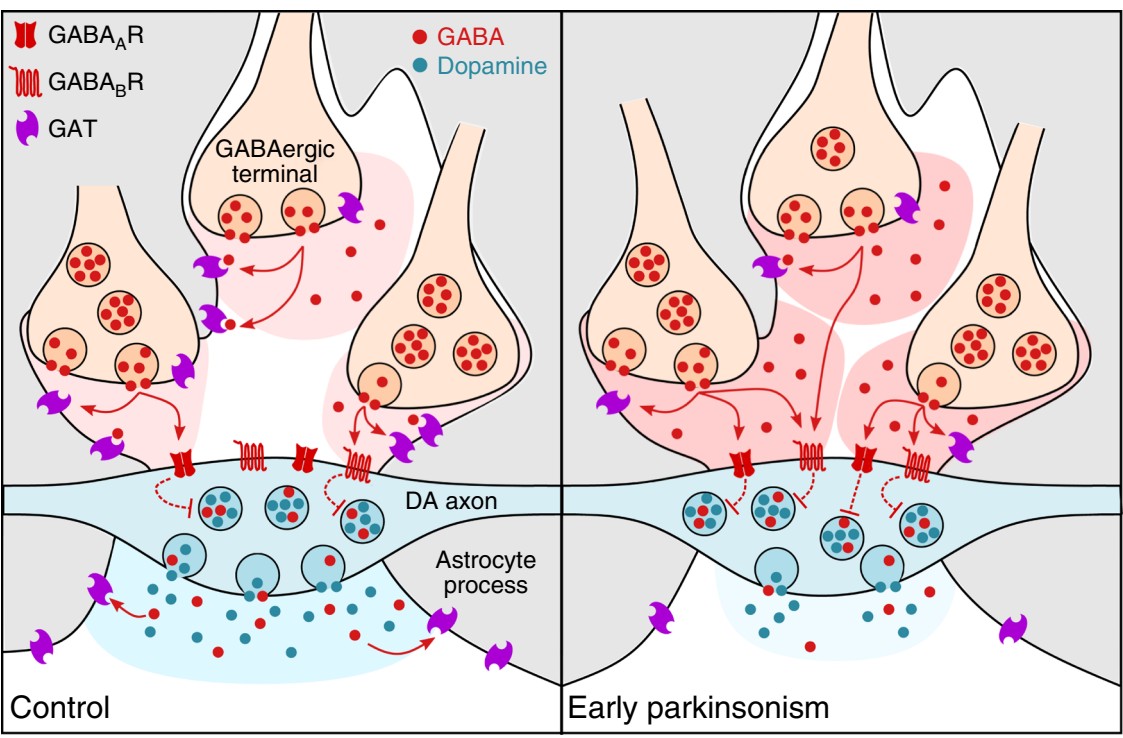

**Fig. 9 Augmented tonic inhibition of striatal DA release in dorsal striatum in early parkinsonism due to reduced striatal GAT expression.** Under normal circumstances (left), GAD-synthesized GABA released from GABAergic striatal neurons can spillover to act at GABA receptors ($GABA_A$R and $GABA_B$R) located presumably on DA axons, inhibiting (dashed red lines) DA and GABA co-release. The level of GABA spillover and tonic inhibition of DA release is determined by the activity of GABA transporters (GATs) located on astrocytes (grey) and neurons, which remove GABA from the extracellular space. In a mouse model of early parkinsonism (right), striatal GAT expression is downregulated in dorsal striatum, resulting in augmented tonic inhibition of DA release by GABA. Co-release of GABA from DA axons is also reduced.

co-release from DA axons, as seen previously[8]. Conversely, ALDH inhibition even slightly boosted tonic GABA inhibition of DA release, suggesting that ALDH-dependent sources of GABA, such as GABA co-release from DA axons, limit the tonic inhibition by the GAD-dependent GABA network. Correspondingly, in mice overexpressing human α-synuclein, in which we found that evoked GABA co-release from DA axons is attenuated, we also found that the levels of tonic GABA inhibition on DA release was boosted. The latter two findings together suggest that deficits in GABA co-release from DA neurons in our PD model might be a driving factor in leading to an enhancement of tonic GABA inhibition on DA axons by the GAD-dependent GABA tone, that in turn further compounds DA and GABA release deficits. We also note that Aldh1a1 mutations in humans and deletion in mice lead to alcohol-consuming preferences[8,49,50], and speculate that

dysregulated DA output might plausibly result and contribute to this behaviour.

The paucity of GABAergic synapses on DA axons[9] suggests that GAD-dependent GABA tone arises from the extrasynaptic ambient tone that can be detected in striatum[11–15]. This tone was action potential independent, i.e. spontaneous[51], as reported previously for tonic inhibition of SPNs[27]. A spontaneous GABAergic regulation of DA release is not surprising when considering that the axonal arbour of a given nigrostriatal DA neuron (in rat) reaches on average 2.7% of the volume of striatum[52,53], and that such volumes contain ~74,000 GABAergic neurons (calculated from 2.8 million striatal neurons per hemisphere[52], of which ~98% are GAD-immunoreactive) and also GAD-positive cholinergic interneurons that can co-release GABA[54]. Even very low rates of spontaneous vesicle release

from a small fraction of GAD-utilizing GABAergic neurons might summate sufficiently to provide a tone at GABA receptors on DA axons that limits DA output. The general functions of this spontaneous GABA tone are not well understood, but could differ from functions of action potential-dependent or synaptic events[10], and could include regulation of DA axonal membrane resistance to modify the impact of other inputs or limit the propagation of action potentials through the axonal arbour for a sparser coding.

We found that GAT-1 and GAT-3 both limit the actions of GABA on DA axons in DLS, and thereby indirectly facilitate DA release. This role for the GATs in supporting DA output was heterogeneous: GATs limited tonic GABAergic inhibition of DA release in DLS, and markedly less so in NAcC, which corresponded with heterogeneity in GAT-1 and GAT-3 expression. Of note, the positive relationship we find between GAT function and DA output is paralleled by, and provides a candidate explanation for, some clinical effects of GAT inhibitors e.g. tiagabine. When used clinically to increase GABA function, anti-epileptics can have parkinsonian-like motor side effects[55].

We were unable to find evidence for robust localization of GAT-1 or GAT-3 proteins to DA axons in DLS, despite previous suggestions that GATs reside on DA axons to support GABA uptake for co-release[7,26]. Because subsequent work has shown that there is a tonic GABAergic inhibition of DA release mediated by both $GABA_A$ and $GABA_B$ receptors[3], we speculate that GATs on other structures might support GABA co-release from DA axons by limiting inhibitory GABA tone, rather than necessarily mediating uptake of GABA into DA axons. Future studies using targeted knockout strategies will be required to assess whether GATs on DA axons support GABA uptake for co-release.

We revealed that astrocytes play a critical role in limiting the tonic GABA inhibition of DA release and therefore supporting DA output. We found that both GAT-3 and, to a lesser extent, GAT-1, could be identified on astrocytes, challenging the long-held generalization that GAT-1 is exclusively neuronal[38]. We show that GATs located on striatal astrocytes in the dorsal striatum provide a major means of limiting GABA inhibition of DA release; our analyses do not exclude similar roles for GATs located on neurons. We found that astrocyte inactivation with the glial metabolic poison fluorocitrate prevented the effects of GAT inhibitors and boosted tonic GABAergic inhibition of DA release. Although the mechanisms and specificity of fluorocitrate are incompletely understood, it remains one of the few tools available to render astrocytes and their transporters inactive. Prolonged treatment with fluorocitrate has the potential to compromise neuronal integrity, but exposure for the short durations used here (1 h) has limited effects on downstream neuron viability[40,41]. The role we find for astrocytes in supporting GABA uptake to limit tonic inhibition of DA release, indicates a previously unappreciated role for astrocytes in regulating the dynamics of DA signaling. This finding significantly revises current understanding of the striatal mechanisms that can dynamically regulate DA transmission. Astrocytic GATs have recently been shown to regulate tonic GABAergic inhibition of striatal SPNs and striatal-dependent behaviors[23], and thus, our collective findings point to GATs and astrocytes as powerful regulators of striatal and DA function that warrant further future investigation.

To probe the wider potential significance of the regulation of striatal DA by striatal GATs, we explored GAT function in a mouse model of early parkinsonism. A recent study in external globus pallidus of dopamine-depleted rodents found elevated extracellular GABA resulting from downregulation of GAT-3 on astrocytes, mediated through a loss of DA signalling at $D_2$ DA receptors[44]. Conversely, striatal GAT-3 levels are upregulated in

mouse models of Huntington's disease[23,27]. In an intriguing parallel seen for glutamate transmission in pre-neurodegenerative β-amyloid-based mouse models of early Alzheimer's disease, hippocampal neurons become hyperactive due to an attenuation of glutamate uptake by astrocytes[56]. Together these emerging strands suggest that impaired astrocyte transporters might be an early underlying feature across neurodegenerative diseases. We explored potential adaptations to GAT function and tonic GABA inhibition of DA release in the striatum of the human α-synuclein-overexpressing mouse model of PD. This model is a highly physiological, slowly progressing mouse model of parkinsonism, that, in capturing a human disease-relevant genetic burden of α-synuclein overexpression, shows early deficits in DA release restricted to dorsal striatum prior to degeneration of DA neurons, disturbed encoding of behaviour of surviving DA neurons and a motor phenotype in old age[32,47]. We firstly ascertained the finding that DA transmission deficits in the DLS of this model in early adulthood are accompanied by a corresponding deficit in GABA co-release from DA axons. Furthermore, we found an augmentation of tonic GABA inhibition of DA release in the DLS (and not NAcC), which was accompanied, and could be explained by, downregulated GAT-1 and GAT-3 expression. It is not yet known whether these adaptations in GAT result from a potential direct interaction between α-synuclein and striatal GATs and/or astrocytes, or whether they are consequential to the reduced dopamine levels, as occurs in astrocytes in globus pallidus after profound depletion of dopamine[44], or to the attenuation of nigrostriatal GABA co-release we observed, which might lead to compensatory lowering of striatal GAT levels. In any event, the resulting enhanced tonic inhibition diminishes nigrostriatal release, compounding the release deficits underpinned by α-synuclein e.g. through tighter vesicle clustering in DA axons[32]. These changes in GATs and tonic GABA inhibition in early parkinsonism can therefore be considered maladaptive to disturbed DA signalling.

In conclusion, the regulation of striatal GABA-DA interactions via striatal GATs and astrocytes represent loci for governing DA output as well as for maladaptive plasticity in early parkinsonism, which could also provide a novel therapeutic avenue for upregulating DA signalling in PD.

## Methods

**Mice**. All procedures were performed in accordance with the Animals in Scientific Procedures Act 1986 (Amended 2012) with ethical approval from the University of Oxford, and under authority of a Project Licence granted by the UK Home Office. Adult (6–8 weeks) wild-type C57BL/6 J mice were obtained from Charles River (Harlow, UK). Knockin mice bearing an internal ribosome entry site (IRES)-linked Cre recombinase gene downstream of the gene *Slc6a3*, which encodes the plasma membrane dopamine transporter, were obtained from Jackson Laboratories (*Slc6a3*[IRES-Cre] mice; *B6.SJL-Slc6a3*[tm1.1(cre)Bkmn]/J; stock no. 006660). PV[Cre] knockin mice expressing Cre recombinase in parvalbumin (PV)-expressing neurons were obtained from Jackson Laboratories (*B6;129P2-Pvalb*[tm1(cre)Arbr]/J; stock no. 008069). SNCA-OVX mice (B6.Cg-Tg(*SNCA*)OVX37Rwm Snca[tm1Rosl]/J; Jackson Laboratories stock no. 023837)[32] are BAC-transgenic mice that over-express human α-synuclein from the *SNCA* genomic locus at Parkinson's disease-relevant levels, and are backcrossed onto a mouse α-synuclein-null (*Snca*[−/−]) background. We made this line optogenetic-capable by crossing with *Slc6a3*[IRES-Cre] mice which were mouse α-synuclein-null (*Slc6a3*[IRES-Cre+/+]; *Snca*[−/−] mice) to generate "SNCA+" mice (*Slc6a3*[IRES-Cre+/−]; *Snca*[−/−]; *SNCA*[+/−]) and "*Snca*[−/−]" littermate control mice (*Slc6a3*[IRES-Cre+/−]; *Snca*[−/−]; *SNCA*[−/−]). Littermate controls were matched for age and sex. All mice were maintained on a C57BL/6 background, group-housed and maintained on a 12-hr light cycle with *ad libitum* access to food and water.

**Stereotaxic intracranial injections**. Slc6a3[IRES-Cre] mice (4–6 weeks), SNCA+ and Snca−/− mice (11–12 weeks) or PV[Cre] mice (4–6 weeks) were anesthetized with isoflurane and placed in a small animal stereotaxic frame (David Kopf Instruments). After exposing the skull under aseptic techniques, a small burr hole was drilled and adeno-associated virus ($8 \times 10^{12}$ genome copies per ml; UNC Vector Core Facility) encoding Cre-dependent ChR2 was injected. Viral solutions were injected at an infusion rate of 100 nL/min with a 32-gauge Hamilton syringe

(Hamilton Company) and withdrawn 5–10 min after the end of injection. In $SNCA+$ and $Snca-/-$ mice, and Slc6a3[IRES-Cre] mice, a total volume of 1 μL of AAV5-EF1α-DIO-hChR2(H134R)-eYFP was injected bilaterally (500 nL per hemisphere/injection) into substantia nigra pars compacta (AP −3.1 mm, ML ± 1.2 mm from bregma, DV −4.25 mm from exposed dura mater). In PV[Cre] mice, a total volume of 600 nL of AAV2-EF1α-DIO-hChR2(H134R)-eYFP was injected bilaterally (300 nL per hemisphere/injection) into dorsolateral striatum (AP+ 0.65 mm, ML ± 2.0 mm from bregma, DV −1.85 mm from exposed dura mater). Viral-injected mice were used for experiments >28 days post-viral injection.

**Slice preparation.** Acute brain slices were obtained from 6–16-week-old mice using standard techniques. Mice were culled by cervical dislocation (for FSCV experiments alone) or mice were anaesthetized with pentobarbital and transcardially perfused with ice-cold artificial cerebrospinal fluid (aCSF) containing (in mM): 130 NaCl, 2.5 KCl, 26 NaHCO$_3$, 2.5 CaCl$_2$, 2 MgCl$_2$, 1.25 NaH$_2$PO$_4$ and 10 glucose (for whole-cell patch-clamp electrophysiology experiments alone or in combination with FSCV experiments). 300-μm thick coronal slices containing striatum were prepared from dissected brain tissue using a vibratome (VT1200S, Leica Microsystems) and transferred to a holding chamber containing a HEPES-based buffer solution maintained at room temperature (20–22 °C) containing (in mM): 120 NaCl, 20 NaHCO$_3$, 10 glucose, 6.7 HEPES acid, 5 KCl, 3.3 HEPES sodium salt, 2 CaCl$_2$, 2 MgSO$_4$, 1.2 KH$_2$PO$_4$ (for FSCV experiments alone) or containing aCSF kept at 34 °C for 15 min before returning to room temperature (20–22 °C) (for whole-cell patch-clamp electrophysiology experiments). All recordings were obtained within 8 h of slicing. All solutions were saturated with 95% O$_2$/5% CO$_2$.

**Fast-scan cyclic voltammetry (FSCV).** Individual slices were hemisected and transferred to a recording chamber and superfused at ~3.0 mL/min with aCSF at 31–33 °C. A carbon fibre microelectrode (CFM; diameter 7–10 μm, tip length 70–120 μm), fabricated in-house, was inserted 100 μm into the tissue and slices were left to equilibrate and the CFM to charge for 30–60 min prior to recordings. All experiments were carried out either in the dorsolateral quarter of the striatum (DLS) or nucleus accumbens (NAc) core (NAcC; within 100 μm of the anterior commissure) or lateral NAc shell (NAcS), one site per slice. Evoked extracellular DA concentration ([DA]$_o$) was measured using FSCV at CFMs as described previously[57]. In brief, a triangular voltage waveform was scanned across the microelectrode (−700 to +1300 mV and back vs Ag/AgCl reference, scan rate 800 V/s) using a Millar Voltammeter (Julian Millar, Barts and the London School of Medicine and Dentistry), with a sweep frequency of 8 Hz. Electrical or light stimuli were delivered to the striatal slices at 2.5 min intervals, which allow stable release to be sustained at ~90–95% over the time course of control experiments. Evoked currents were confirmed as DA by comparison of the voltammogram with that produced during calibration with applied DA in aCSF (oxidation peak +500–600 mV and reduction peak −200 mV). Currents at the oxidation peak potential were measured from the baseline of each voltammogram and plotted against time to provide profiles of [DA]$_o$ versus time. CFMs were calibrated post-hoc in 2 μM DA in each experimental solution. Calibration solutions were made immediately before use from stock solution of 2.5 mM DA in 0.1 M HClO$_4$ stored at 4 °C. CFM sensitivity to DA was between 10 and 40 nA/μM. Unless noted otherwise, FSCV recordings were carried out in the presence of dihydro-β-erythroidine (DHβE, 1 μM), an antagonist at β2 subunit-containing nicotinic acetylcholine receptors (nAChRs), to eliminate cholinergic signalling effects on DA release[57–59]. Release was tetrodotoxin-sensitive as shown previously[57].

In experiments where [DA]$_o$ was evoked by electrical stimulation, a local bipolar concentric Pt/Ir electrode (25 μm diameter; FHC Inc.) was placed ~100 μm from the CFMs and stimulus pulses (200 μs duration) were given at 0.6 mA (perimaximal in drug-free control conditions). We applied either single pulses (1p) or 2–10 pulses (2p, 4p, 5p, and 10p) at 10–100 Hz. A frequency of 100 Hz is useful as a tool for exposing changes in short-term plasticity in DA release that arise through changes in initial release probability[37,59]. In experiments where [DA]$_o$ was evoked by light stimulation in slices prepared from Slc6a3[IRES-Cre] mice expressing ChR2, DA axons in striatum were activated by TTL-driven (Multi Channel Stimulus II, Multi Channel Systems) brief pulses (2 ms) of blue light (470 nm; 5 mWmm$^{-2}$; OptoLED; Cairn Research), which illuminated the field of view (2.2 mm, ×10 water-immersion objective). Epifluorescence (520 nm) used to visualize ChR2-eYFP expression was used sparingly to minimize ChR2 activation before recordings.

**Electrophysiology.** Individual slices were hemisected and transferred to a recording chamber and superfused at ~3.0 mL/min with aCSF at 31–33 °C. Cells were visualized through a ×40 water-immersion objective with differential interference contrast optics. All whole-cell experiments were recorded using borosilicate glass pipettes with resistances in the 3–5 MΩ range and were pulled on a Flaming-Brown micropipette puller (P-1000, Sutter Instruments). Whole-cell voltage-clamp electrophysiology recordings were made from spiny projection neurons (SPNs; identified by their membrane properties[60,61]) in the DLS or NAcC. SPNs were voltage clamped at −70 mV using a MultiClamp 700B amplifier (Molecular Devices) and with pipettes filled with a CsCl-based internal solution (in mM 120

CsCl, 15 CsMeSO$_3$, 8 NaCl, 0.5 EGTA, 10 HEPES, 2 Mg-ATP, 0.3 Na-GTP, 5 QX-314; pH 7.3 adjusted with CsOH; osmolarity ranging from 305–310 mOsmkg$^{-1}$). The recording perfusate always contained NBQX (5 μM) and APV (50 μM) to block AMPA and NMDA receptor-mediated inward currents. Errors due to the voltage drop across the series resistance (<20 MΩ) were left uncompensated and membrane potentials were corrected for a ~5 mV liquid junction potential. Cells were discarded from analysis if if series resistance varied by more than 15% or increased over 25 MΩ.

To record tonic GABA$_A$ currents, SPNs voltage clamped at −70 mV were recorded in gap-free mode. Cells were allowed to stabilize for 5–10 min before drug manipulations: GAT inhibitors were bath applied for 20–25 min; picrotoxin (100 μM) for an additional 3–5 min. Recordings of light-evoked GABA currents in SPNs from ChR2-expressing DA axons in slices from Slc6a3[IRES-Cre], $SNCA+$ and $Snca-/-$ mice were taken 10 min after break-in, and at 30 s intervals for a duration of 10 min from SPNs voltage clamped at −70 mV. Under these conditions, GABA$_A$ receptor-mediated currents appear inward as reported previously[6]. TTL-driven (Multi Channel Stimulus II, Multi Channel Systems) brief pulses (2 ms) of blue light (470 nm; 5 mWmm$^{-2}$; OptoLED; Cairn Research) illuminated the full field of view (2.2 mm, ×10 water-immersion objective).

**High-performance liquid chromatography.** Tissue DA content was measured by HPLC with electrochemical detection in tissue punches from dorsal and ventral striatum as described previously[32]. Tissue punches from the dorsal striatum (2.0 mm diameter, aligned to the edge of the corpus callosum on the dorsolateral side) and ventral striatum (1.5 mm diameter, centered on the anterior commissure) from two brain slices per animal were taken and stored at −80 °C in 200 μL 0.1 M HClO$_4$. On the day of analysis, samples were thawed on ice, homogenized, and centrifuged at 15,000 × $g$ for 15 min at 4 °C. The supernatant was analysed for DA content. Analytes were separated using a 4.6 × 250 mm Microsorb C18 reverse-phase column (Varian or Agilent) and detected using a Decade II SDS electrochemical detector with a Glassy carbon working electrode (Antec Leyden) set at +0.7 V with respect to a Ag/AgCl reference electrode. The mobile phase consisted of 13% methanol (vol/vol), 0.12 M NaH$_2$PO$_4$, 0.5–4.0 mM octenyl succinic anhydride (OSA) and 0.8 mM EDTA (pH 4.4–4.6), and the flow rate was fixed at 1 mL/min. Analyte measurements were normalized to tissue punch volume (pmol/mm$^3$). HPLC data was collected with Clarity (DataApex).

**Western blot.** Mouse brains were extracted and sliced using the procedures outlined above. One 1.2-mm thick coronal slice containing striatum was prepared from each brain and one tissue punch (2 mm in diameter) of dorsal striatum taken per hemisphere. Striatal tissue samples were snap frozen and stored at at −80 °C. For analysis, striatal tissue was defrosted on ice, homogenized in RIPA Lysis and Extraction Buffer (Sigma) containing 150 mM NaCl, 1.0% IGEPAL, 0.5% sodium deoxycholate, 0.1% SDS, 50 mM Tris, pH 8.0, with Complete-Mini Protease Inhibitor and PhosStop (Roche), using a Tissue Tearor (Biospec Products, Inc) and soluble fraction isolated by microcentrifugation at 15,000 × $g$ for 15 min at 4 °C. Total protein content was quantified using a BCA Protein Assay Kit (Thermo Scientific) and equal amounts of total protein were loaded onto 4–15% Tris-Glycine gels (BioRad). Following electrophoresis (200 V for ~45 min), proteins were transferred onto polyvinylidene fluoride membranes (BioRad). Blots were probed overnight at 4 °C with 1:1000 rabbit anti-GABA transporter 1 (Synaptic Systems, 274102) or 1:1000 rabbit anti-GABA transporter 3 (Abcam, AB181783). Blots were incubated with HRP-conjugated secondary anitbodies at 1:3000 for 1 h at room temperature and bands developed using ECL Prime Western Blotting Detection Reagent (GE Healthcare). Blots were subsequently incubated with 1:20000 HRP-conjugated β-actin (Abcam, AB49900) for 1 h at room temperature and bands developed as above. Visualization and imaging of blots was performed with a ChemiDoc Imaging System (BioRad) and bands quantified using Image Lab Software (version 5.1, BioRad). Protein concentration for GAT-1 and GAT-3 were normalized to β-actin before normalizing to $Snca-/-$ controls. Uncropped blots from which the data were derived are provided in the Source Data file.

**Indirect immunofluorescence.** Mice were anaesthetized with an overdose of pentobarbital and transcardially perfused with 20–50 mL of phosphate-buffered saline (PBS), followed by 30–50 mL of 4% paraformaldehyde (PFA) in 0.1 M phosphate buffer, pH 7.4. Brains were removed and post-fixed overnight in 4% PFA. Brains were embedded in agar (3–4%) and coronal sections (50 μm) were cut on a vibrating microtome (Leica VT1000S) and collected in a 1 in 4 series. Sections were stored in PBS with 0.05% sodium azide. Upon processing, sections were washed in PBS and then blocked for 1 h in a solution of PBS TritonX (0.3%) with sodium azide (0.02%; PBS-Tx) containing 10% normal donkey serum (NDS). Sections were then incubated in primary antibodies overnight in PBS-Tx with 2% NDS at 4 °C. Primary antibodies: rabbit anti-TH (1:2000, Sigma–Aldrich, ab112); rabbit anti-GAT-1 (1:1000, Synaptic Systems, 274102); rabbit anti-GAT-3 (1:250, Millipore/Chemicon, AB1574); rabbi anti-NeuN (1:500, Biosensis, R-3770–100); guinea pig anti-S100β (1:2000, Synaptic Systems, 287004); rat anti-GFP that also recognizes eYFP (1:1000, Nacalai Tesque, 04404-84) and guinea pig anti-parvalbumin (1:1000, Synaptic Systems, 195004). Sections were then incubated in species-appropriate fluorescent secondary antibodies with minimal cross-reactivity overnight in PBS-Tx at room

temperature (Donkey anti-Rabbit AlexaFluor 488, 1:1000, Invitrogen, A21206; Donkey anti-Rabbit Cy3, 1:1000, Jackson ImmunoResearch, 711-165-152; Donkey anti-Guinea Pig AlexaFluor 488, 1:1000, Jackson ImmunoResearch, 706-545-148; Donkey anti-Rat AlexaFluor 488, 1:1000, Jackson ImmunoResearch, 712-545-153). Sections were washed in PBS and then mounted on glass slides and cover-slipped using Vectashield (Vector Labs). Coverslips were sealed using nail varnish and stored at 4 °C. To verify the specificity of ChR2-eYFP expression in TH-positive midbrain structures in Slc6a3$^{IRES-Cre}$ mice (see Fig. 1a), mounted sections were imaged with an Olympus BX41 microscope with Olympus UC30 camera and filters for appropriate excitation and emission wave lengths (Olympus Medical).

**Confocal imaging and image analysis**. Confocal images were acquired with an LSM880/Axio.Imager Z2 (Zeiss) and Image J was used for image analysis. For whole striatum analysis of GAT-1 or GAT-3, the ×10 (NA = 0.45) objective was used and all imaging settings (laser %, pinhole/optical section, pixel size, gain, and scanning speed) were kept constant between animals. For the quantification of fluorescence (mean grey values), four sections in the rostro-caudal plane were imaged at approximately the following distances rostral of Bregma; +1.3, +1.0, +0.6 and +0.25 mm (see Supplementary Fig. 5). A region of interest (ROI) of 300 × 300 μm was overlaid over the DLS and the ventral caudate putamen (vCPu); and an ROI of 200 × 200 μm was overlaid on NAcC and the NAcS, for both hemispheres. Values for NAcC and NAcS were taken from the two most rostral sections (see Supplementary Fig. 5). Mean grey values from the areas of interest were normalized to the median grey value for each hemisphere ($n = 12$ hemispheres from 6 animals). For examination of colocalization a ×63 objective was used (NA = 1.46); Z-stacks were taken with the pinhole set to 1 Airy Unit (optical section = 0.7 μm) with a z-stack interval of 0.30 μm or 0.35 μm. In order to assess colocalization ZEN (blue edition v.2.3; Zeiss) software was used. For S100β, PV + axons (eYFP in PV$^{Cre}$ mice) or DA axons (eYFP in Slc6a3$^{IRES-Cre}$ mice) and GAT-1 or GAT-3 colocalization, stacks from a minimum of two striatal regions and two NAcC regions in at least one section were examined per animal ($n = 3$ per marker).

**Drugs**. (S)-SNAP5114 (SNAP, 50 μM), (±)-nipecotic acid (NPA, 1.5 mM), 3-mercaptopropionic acid (3-MPA, 500 μM), γ-aminobutyric acid (GABA, 2 mM) and picrotoxin (100 μM) were obtained from Sigma–Aldrich. Dihydro-β-erythroidine hydrobromide (DHβE, 1 μM), (+)-bicuculline (10 μM), (S)-MCPG (200 μM) and tetrodotoxin (TTX, 1 μM) were obtained from Tocris Bioscience. DL-2-Amino-5-phosphonovaleric acid (AP5, 50 μM), disulfiram (10 μM) and SKF89976A hydrochloride (SKF, 20 μM) were obtained from Santa Cruz Biotechnology. NBQX disodium salt (NBQX, 5 μM) and CGP 55845 hydrochloride (CGP, 4 μM) were obtained from Abcam. Fluorocitrate was prepared as previously described[39]. In brief, D,L-fluorocitric acid Ba$_3$ salt (Sigma–Aldrich) was dissolved in 0.1 M HCl, the Ba$^{2+}$ precipitated with 0.1 M Na$_2$SO$_4$ and then centrifuged at $1000 \times g$ for 5 min. Supernatant containing fluorocitrate was used at a final concentration of 200 μM for experimentation. All drugs were dissolved in distilled water or dimethyl sulfoxide (DMSO) to make stock aliquots at 1000–10,000× final concentrations and stored at −20 °C. Stock aliquots were diluted with aCSF to final concentration immediately before use.

**Data acquisition and analysis**. FSCV data were digitized at 50 kHz using a Digidata 1550 A digitizer (Molecular Devices). Data were acquired and analyzed using Axoscope 11.0 (Molecular Devices) and locally written VBA scripts in Microsoft Excel (2013). For drug effects, peak [DA]$_o$ was averaged over four stimulations once peak [DA]$_o$ had restabilized post-drug application and compared to time-matched data from drug-free controls, unless otherwise stated. We observed modest run-down in [DA]$_o$ evoked by 1p electrical stimulations in slices preincubated in fluorocitrate (200 μM) and therefore we used an alternative stimulation paradigm to compare a large number of dorsal striatal recording sites in slices pretreated with fluorocitrate versus control conditions to minimize run-down. There are previous reports of a fluorocitrate-dependent slow run-down of excitatory postsynaptic currents in the hippocampus[42], possibly reflecting astroglia's role in stable synaptic neurotransmission.

FSCV data were normalized to predrug conditions for clarity and for comparisons between regions. For experiments involving multiple pulse protocols, each stimulation type was repeated in triplicate, interspersed with 1p stimulations, and then averaged and normalized to 1p stimulations at each recording site, as previously[3,57].

Membrane currents from voltage-clamp electrophysiology experiments were amplified and low-pass filtered at 5 kHz using a MultiClamp 700B amplifier (Molecular Devices), digitized at 10 kHz and acquired using a Digidata 1550 A digitizer (Molecular Devices). Peak amplitude, onset latency, peak latency, 10–90% rise time and decay time were measured from an average of three replicate traces recorded before and after drug wash on conditions using Clampfit 10.4.1.4 software (Molecular Devices).

For all experiments, data were collected from a minimum of three animals. Data were compared for statistical significance using Prism 7 (Graph Pad) with the following statistical tests (as indicated in the text, and two-tailed): unpaired $t$-tests, paired $t$-tests, two-way repeated-measures ANOVA followed by Sidak's multiple comparison tests, and where the data were not normally distributed, Mann–Whitney U tests, Kruskal–Wallis ANOVA followed by Dunn's Multiple Comparisons, Friedman's ANOVA on Ranks and Student–Newman–Keuls multiple comparisons and for comparing cumulative distributions, Komogorov–Smirnov tests. $p$ values smaller than 0.05 were considered statistically significant, adjusted for multiple comparisons.

**Reporting summary**. Further information on research design is available in the Nature Research Reporting Summary linked to this article.

## Data availability
The authors declare that all datasets generated and analysed during this study are available within this paper and its Supplementary Files. The source data underlying all figures and supplementary figures are provided as a Source Data file.

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

## Acknowledgements

This work was supported by grants from Parkinson's UK (G-1504, and Monument Trust Discovery Award J-1403), a Clarendon Fund Studentship awarded to B.M.R. and a Fundação para a Ciência e a Tecnologia studentship awarded to E.F.L. N.M.D. and P.J.M. were supported by the UK Medical Research Council (Awards MC_UU_12024/2 and MC_UU_00003/5 to P.J.M.) and the Wellcome Trust (Investigator Award 101821 to P.J. M.). We thank Ben Micklem and Lisa Conyers for their assistance with anatomical work, Drs Yoland Smith and Jean-Francois Pare for GAT antibody recommendations, Milena Cioroch for technical expertise in maintaining transgenic mouse colonies, and members of the Cragg laboratory for discussions throughout the course of this study.

## Author contributions

B.M.R. and S.J.C. conceived and designed the research, and wrote the manuscript with input from all other authors. B.M.R. performed the majority of experiments and analysis of data. N.M.D. and P.J.M. collected and interpreted anatomical confocal imaging data. K.R.B., E.F.L., S.T., R.E.S. and N.P. assisted with the collection, analysis and interpretation of voltammetry data. N.C.-R. and N.B.-V. provided expertise and assistance with western blot experiments. R.W.-M. conceived of and provided *SNCA*-OVX mice and assisted in interpretation of data.

## Competing interests

The authors declare no competing interests.
