## [Peer Review File · Nature Communications]

Reviewers' Comments:

Reviewer #1:

Remarks to the Author:

Roberts et al. address the mechanisms of dopamine (DA) release in the striatum. The problem is important since DA release is suppressed in Parkinson's. The authors have previously published that DA antagonists of GABA receptors stimulate DA release in the striatum, leaving open the question of the mechanism. In this report they go a step further and compare dorsal and ventral regions of the striatum. Using Voltammograms and optogenetic approach in Parkinson mouse models, they report that DA release is modulated by GAT 1 and 3, and regulation is suppressed in SL6a3 mouse model for Parkinson. The strength of the manuscript is the beautiful technical approach and elegant use of mouse models and optogenetic models. The experiments are nicely performed, the results are consistent, and support the conclusions of the paper.

A few weaknesses or clarifications should be addressed by the authors. The manuscript, although very carefully performed, is limited to cell recording and the use of pharmacological agents, which can have unintended effects. It would be strong if the authors could use at least one other technique to solidify the conclusions. For example, siRNA knockdown would be very satisfying to test at least some of their ideas. For example, the authors use a specific inhibitor for GAT1 and compare the results for DA release to a combined inhibition of GAT1 and GAT3 inhibitors. It might be difficult to discriminate fractional differences of inhibition. Might not siRNA be a strong discriminator between GAT1 and GAT3 and their cell type effects? THsi seems possible since in (Fig. 1C), the authors inject AAVCHR2 YFP to test DA levels and siRNA could also be introduced by the same method. The authors should discuss and clarify how their analyses differ from an article published around a year ago. (GABA transporters (GAT3) regulates DA release in astrocytes of rats, 2018). The authors need to clarify what new information is presented here

Reviewer #2:

Remarks to the Author:

Roberts and colleagues have submitted a very nice and detailed paper that explores the mechanisms that govern dopamine (DA) release in the dorsal and ventral striatum. They have focussed on the role of ambient or tonic GABA and uncovered important roles of GAT1 and GAT3 transporters in these effects. Both of these transporters are found on astrocytes and display interesting differences in expression between dorsal and ventral regions.

Overall, this is a carefully performed and detailed study that is reported very nicely with a great deal of clarity. It takes the logical next step from recent RNA-seq work that showed astrocyte GAT1 and GAT2 mRNA expression, and shows functional effects. This alone is an important contribution, but the authors go one step further and show downregulation of GATs in a mouse model of Parkinson's disease.

I have no major concerns with this manuscript and feel as though it could be published as is. It reports exciting new data that are much needed. It will appeal to a diverse audience of researchers working on astrocytes, PD and neural circuits. The following points could be easily addressed in a revision.

1. I found some of the figures to be a little confusing. At first I could not determine why, but eventually I realized it was because of two things. First, in many figures the authors have embedded panels into panels. For example in Fig 1 and 2, all of the figures showing time courses have embedded traces. I think this makes the figures hard to interpret and perhaps the authors would consider revising the figures to place the traces next to, rather than within, the graphs showing time courses. Second, there is a lot of use of gray bars to show when an effect occurs in most figures. I found this distracting. Perhaps this type of shading in the figures could be reduced?

This is especially the case in Figure 3 where brown shading is used.

2. In some cases attention is needed to the labeling of figure panels. For example, in Fig 5 purple font is used on a purple image. Other figures require similar attention.

3. Although the use of fluorocitrate is necessary, the authors should discuss the caveats associated with its use. It is essentially a metabolic poison that can not be assumed to be astrocyte selective. This can be addressed by simply discussing the limitations of the pharmacological approach, which reflects the current state of the field.

4. Some of the fonts are very small. For example, in Figure 4.

Overall, a very nice paper that requires some simple revisions.

Reviewer #3:

Remarks to the Author:

This study builds upon a previous one from the same group (Lopes et al., 2018), which revealed presynaptic modulation of dopamine (DA) release from midbrain axons in striatum by GABA using direct pharmacological manipulation of GABA_A and GABA_B receptors. Here, Roberts and colleagues now show that endogenous elevation of GABA in striatal slices following pharmacological block of the neuronal and astrocytic membrane GABA transporters GAT1 and GAT3 also depresses DA release in dorsolateral striatum (DLS) more so than in nucleus accumbens (NAc). The authors go on to show that most GAT proteins in striatum distribute to astrocytes in DLS and that pharmacological poisoning of astrocytes with fluorocitrate occludes the inhibition of DA release by the non-selective GAT blocker NPA, pointing to astrocytes as the main source of extrasynaptic GABA produced by broad pharmacological antagonism of GAT in DLS. They then make the interesting observation that in a mouse model of Parkinson's disease, prior to overt neurodegeneration, DA and GABA release from midbrain dopaminergic axons is decreased in DLS as a result of excessive buildup of GABA in striatum, possibly caused by downregulation of striatal GAT1 and GAT3 protein levels.

Overall, this study makes a compelling case for endogenous accumulation of GABA being capable of exerting a negative modulatory action on synaptic release of DA and GABA from midbrain dopaminergic axons, and for this mechanism contributing to diminished DA signaling in early parkinsonism. That said, many of this study's findings repeat prior published reports or lack experimental support, making it not suitable for publication in Nature Communications.

MAJOR COMMENTS:

1. Novelty: Significant sections of this manuscript repeat previous findings already published by this group and others. While this is generally a good thing, it does leave this study relatively light on original material. Specifically, the authors had already published a study (Lopes et al. 2018) showing that GABA inhibits electrically and optogenetically evoked DA release in DLS independent of local cholinergic signaling, which is now repeated in significant sections of Figure 1 and in Fig 2K. A logical extension of this previous work is that other means of elevating GABA receptor signaling in slice ought to have comparable effects, including block of the membrane GABA transporters GAT1 and GAT3, which have been extensively shown elsewhere to increase extracellular GABA levels in striatum (Ade et al. 2008; Cepeda et al 2013; Kirmse et al 2008; Santhakumar et al 2010). The findings in Figure 2 are therefore not surprising, and those in Figure 3 merely replicate of those previous studies.

2. Regional differences: This manuscript reveals that tonic GABA signaling in slice hampers DA release in both DLS and NAc (Figure 1), but that only DLS is susceptible to further increases in presynaptic inhibition by GABA following NPA treatment (a non-selective antagonist of GAT1 and GAT3; Figure 2). They conclude that GAT does not significantly limit extrasynaptic GABA levels in NAc, and support this claim using immunohistochemistry (Figure 4), showing a slight but significant "enrichment" in DLS relative to NAc for GAT1 and GAT3. This figure is highly deceiving,

as the authors normalized GAT immunofluorescence within striatum to min/max values. While this can be used to reveal slight differences, it cannot be used to comment on the magnitude or biological relevance of the said difference. Moreover, the histograms in panels B and D only span 0.8-1.2 a.u. to further accentuate small differences. Consistent with this, the authors show in Figure 3 that NPA doubles extrasynaptic GABA tone in NAc. Thus, GATs do limit GABA accumulation in NAc, and the absence of presynaptic modulation of GABA release by NPA revealed earlier (Figure 2) ought to be attributed to something other than GATs. This point is currently not made clear, and their conclusion of an effect specific to DLS is stretched, at best. Importantly, previous studies reported differences in the ability of direct and indirect pathway MSNs to detect extrasynaptic/tonic GABA (Ade et al 2008; Santhakumar et al 2010); because this study does not distinguish direct and indirect pathway MSNs and because their sample data is exceedingly small (N=6), the small difference in NAc and DLS may reflect a sampling bias rather than true areal differences. In addition, because extracellular GABA levels are not measured directly (but rather via a standing GABAergic conductance in MSNs), the effect reported in Figure 3C may not reflect differences in extracellular GABA as much as differences in the sensitivity of MSNs of both pathways to extrasynaptic/tonic GABA in DLS vs. NAc. Together, these concerns mitigate the significance of the major point made in Figures 1 through 4.

3. Assigning GAT effects to striatal astrocytes: Immunofluorescence is used to localize GAT protein to astrocytes, but the evidence provided is underwhelming. First, existing gene and protein expression data already make a compelling case for it. Second, in Figure 5A-B, the authors conclude that antibodies to S100B can be successfully used to label striatal astrocytes, but no counterstain is provided to support their argument. What if some fraction of labeled somata were neurons? Third, their co-stain with antibodies to GAT1 and GAT3 in Fig 5C-D is amongst the least convincing I have seen. GAT fluorescence is not seen within S100B+ cell bodies, and instead looks to be distributed in fine processes that are adjacent to S100B+ processes, with little to no overlap. This is more consistent with astrocytic processes enveloping GAT+ axonal varicosities. Fourth, no quantification is provided. Thus, the only evidence pointing to astrocytes as being the main target of NPA and contributors of tonic GABA is the fluorocytrate experiment, whose effects on neurons cannot be ruled out. The authors should therefore refrain from constantly referring to extracellular GABA as being exclusively astrocyte-derived, they should rephrase their title and update their discussion to acknowledge this point.

4. Absence of GAT molecules in dopamine axons: Although peripheral to their main story, the authors make a point to challenge a previous finding that co-release of GABA from DA axons requires GAT-mediated presynaptic uptake of GABA (Tritsch et al 2014). This earlier study used GAT blockers and reported a near complete loss of GABA co-release. The present study finds that GAT blockade depresses DA release by 30-40% via presynaptic GABA receptors, indicating that some presynaptic inhibition may have contributed to the co-release phenotype. However, the Tritsch et al study controlled for such confound by using other means of elevating GABA tone and by limiting presynaptic modulation with GABA_B receptor antagonists. Nevertheless, Roberts and colleagues suggest that DA axons do not express GAT (challenging the mechanism championed by Tritsch et al) using immunohistochemistry, which is inherently limited in its ability to categorically rule out protein expression, as the technique can only provide relative comparisons given experimental variables like antibody sensitivity and a microscope's imaging settings. If the authors insist on challenging the results of the Tritsch et al. (2014) study, they ought to do so directly, perhaps by repeating their experiments in GABA_B antagonists and showing that any means of increasing extracellular GABA in striatum (like fluorocytrate treatment) are sufficient to abolish GABA co-release from DA axons. If on the other hand the authors are not willing to do so directly and thoroughly, they ought to keep it to a discussion point.

5. Figure 6 describes a set of interesting observations in a mouse model of Parkinson's disease that superficially support their model, but several key experiments well within the reach of the experimenters are missing to more conclusively connect the dots. First, the authors do not exclude the possibility that neuronal loss contributed to the observed decrease in DA and GABA release from SNc axons. They merely cite previous work, but the experimental conditions here are different, as the authors bred this mutation to Slc6a3-Cre mice. Second, the authors report decreased GAT expression in striatum by western blot, but this decrease cannot be ascribed to

astrocytes, as their conclusions consistently suggest. Third, the authors do not directly demonstrate that extracellular levels of GABA are actually elevated in SNCA+ mice. Recordings similar to those shown in figure 3 are warranted. Fourth, does fluorocytate occlude the effect of GABAR antagonists on DA shown in Figure 6H? Fifth, is GABA co-release in Figure 6E also subject to presynaptic inhibition at baseline. In other words, does application of a GABA_B receptor antagonist potentiate GABA co-release? If so, does applying a GABA_B receptor antagonist to SNCA+ mice restore IPSCs to the same amplitude as controls? If not, it would be highly suggestive of additional mechanisms taking place, which ought to be acknowledged.

MINOR COMMENTS

1. The authors seem to assume that ALDH1a1 is only expressed in DA neurons and that disulfiram selectively blocks GABA synthesis in these cells. However, ALDH1a1 is abundantly expressed throughout the brain by astroglial cells (<https://www.proteinatlas.org/ENSG00000165092-ALDH1A1/tissue/mouse+brain>), and ALDH1a1's function is not limited to producing GABA. Moreover, disulfiram is far from a selective ALDH1a1 antagonist. This manipulation will therefore impact many dehydrogenases throughout the brain, possibly confounding the author's interpretation of the data.
2. In Fig. 1E, the authors rely on baseline-normalized data to show that GABA receptor antagonists have a greater effect on DA release in the presence of disulfiram compared to control. It is however possible that one of disulfiram's many effects is to elevate tonic GABA levels, and thus depress DA release more strongly at baseline. The authors ought to present and statistically compare raw values before bath application of GABAR antagonists to exclude this possibility. The authors also ought to repeat the measurements of tonic GABA in MSNs (as in Figure 3).
3. For experiment comparing before and after drug application, the authors ought to use a Wilcoxon signed rank test (not Mann-Whitney). Such comparison may reveal a statistically significant effect of NPA on fluorocytate-treated slices in Figure 5G.

Response to Reviewers

Reviewers comments are in black italics. Our responses are in blue.

Reviewer #1 (Remarks to the Author):

Roberts et al. address the mechanisms of dopamine (DA) release in the striatum. The problem is important since DA release is suppressed in Parkinson's. The authors have previously published that DA antagonists of GABA receptors stimulate DA release in the striatum, leaving open the question of the mechanism. In this report they go a step further and compare dorsal and ventral regions of the striatum. Using Voltammograms and optogenetic approach in Parkinson mouse models, they report that DA release is modulated by GAT 1 and 3, and regulation is suppressed in SL6a3 mouse model for Parkinson. The strength of the manuscript is the beautiful technical approach and elegant use of mouse models and optogenetic models. The experiments are nicely performed, the results are consistent, and support the conclusions of the paper.

A few weaknesses or clarifications should be addressed by the authors. The manuscript, although very carefully performed, is limited to cell recording and the use of pharmacological agents, which can have unintended effects. It would be strong if the authors could use at least one other technique to solidify the conclusions. For example, siRNA knockdown would be very satisfying to test at least some of their ideas. For example, the authors use a specific inhibitor for GAT1 and compare the results for DA release to a combined inhibition of GAT1 and GAT3 inhibitors. It might be difficult to discriminate fractional differences of inhibition. Might not siRNA be a strong discriminator between GAT1 and GAT3 and their cell type effects? This seems possible since in (Fig. 1C), the authors inject AAVCHR2 YFP to test DA levels and siRNA could also be introduced by the same method.

RESPONSE: We agree that it is very important to be wary of off-target effects of pharmacological agents. We controlled for the potential off-target effects of GAT inhibitors in several ways. We conducted experiments using GAT inhibitors with co-application of GABA receptor antagonists which confirm that GAT inhibitors are acting by boosting GABA action (**Fig. 2** and **Fig. 3** in revised manuscript). Furthermore, we used a range of GAT inhibitors (**Fig. 2** in revised manuscript), which all exerted the same kinds of effects. These experiments ensured that pharmacological GAT inhibitors exerted their effects on DA release through preventing GABA uptake from the extracellular environment and increasing GABA receptor tone in the striatum. In our revised manuscript, we have **included new data** to provide evidence that GAT inhibition does not attenuate DA release through other mechanisms that might regulate DA, including D₂ receptors, ionotropic and metabotropic glutamate receptors, and DA uptake transporters (**Supplementary Fig. 3**).

We did not intend to indicate that either of the GAT isoforms (GAT-1 vs GAT-3) is more influential than the other in limiting GABA receptor-mediated inhibition of striatal DA release, but rather, we suggest that both isoforms play important roles. In our revised manuscript, we have **included new complementary data** that also shows the effect of GAT-3 inhibition alone, as well as in the prior presence of GAT-1 inhibition (**Fig. 2** in revised manuscript). Our results show that a GAT-3 role can be readily detected but that its role is underestimated when GAT-1 can compensate for its impairment, and more strongly exposed when GAT-1 is also blocked.

An siRNA knockdown to test for GAT isoform-specific effects has its own limitations: not only is it a surprisingly lengthy process to develop (from scratch) and validate (with numerous controls for vectors and constructs), but given the strong role for two isoforms and their potential compensatory responses, siRNA knockdown of one isoform might very likely result in rapid and sustained compensatory upregulation of the other isoform, or have differential efficacy for different constructs, making comparisons difficult.

The authors should discuss and clarify how their analyses differ from an article published around a year ago. (GABA transporters (GAT3) regulates DA release in astrocytes of rats, 2018). The authors need to clarify what new information is presented here.

RESPONSE: We cannot find an article matching that name or description, unless the reviewer is referring to Chazalon et al., (2018) “GAT-3 Dysfunction Generates Tonic Inhibition in External Globus Pallidus Neurons in Parkinsonian Rodents”. Cell Reports 23, 167-1960, which we already describe and reference many times throughout our manuscript (and noting the Chazalon paper details GABA transmission and transporters in the globus pallidus, not the striatum).

Reviewer #2 (Remarks to the Author):

Roberts and colleagues have submitted a very nice and detailed paper that explores the mechanisms that govern dopamine (DA) release in the dorsal and ventral striatum. They have focussed on the role of ambient or tonic GABA and uncovered important roles of GAT1 and GAT3 transporters in these effects. Both of these transporters are found on astrocytes and display interesting differences in expression between dorsal and ventral regions.

Overall, this is a carefully performed and detailed study that is reported very nicely with a great deal of clarity. It takes the logical next step from recent RNA-seq work that showed astrocyte GAT1 and GAT3 mRNA expression, and shows functional effects. This alone is an important contribution, but the authors go one step further and show downregulation of GATs in a mouse model of Parkinson's disease.

I have no major concerns with this manuscript and feel as though it could be published as is. It reports exciting new data that are much needed. It will appeal to a diverse audience of researchers working on astrocytes, PD and neural circuits. The following points could be easily addressed in a revision.

1. I found some of the figures to be a little confusing. At first I could not determine why, but eventually I realized it was because of two things. First, in many figures the authors have embedded panels into panels. For example in Fig 1 and 2, all of the figures showing time courses have embedded traces. I think this makes the figures hard to interpret and perhaps the authors would consider revising the figures to place the traces next to, rather than within, the graphs showing time courses. Second, there is a lot of use of gray bars to show when an effect occurs in most figures. I found this distracting. Perhaps this type of shading in the figures could be reduced? This is especially the case in Figure 3 where brown shading is used.

RESPONSE: We thank the reviewer for raising their issues with the figures. In our revised manuscript, we have now **revised all the figures** as suggested in order to (i) to re-position the DA traces next to the graphs rather than embedding within, and (ii) to reduce the shading across all figures.

2. In some cases attention is needed to the labelling of figure panels. For example, in Fig 5 purple font is used on a purple image. Other figures require similar attention.

RESPONSE: We have **revised the figures** to help clarify. For example, we have lightened the referenced text on our immunohistochemistry figures, but have chosen to keep the hue of the colour the same because it indicates which protein is represented by each colour channel. We hope that we have made the text more discernible from the confocal images displayed behind.

3. Although the use of fluorocitrate is necessary, the authors should discuss the caveats associated with its use. It is essentially a metabolic poison that can not be assumed to be astrocyte selective. This can be addressed by simply discussing the limitations of the pharmacological approach, which reflects the current state of the field.

RESPONSE: We thanks the reviewer for this constructive feedback and agree that a discussion of its use would be a good addition. We have **revised the text** to discuss this approach in more detail. As the reviewer notes, there are very limited methods available for inactivating astrocytes and their transporters. We have now added a fuller description in support of its use in the Results (P7), and discussion the caveats in the Discussion (P11). These sections now read:

P7 *“We exposed striatal slices to the gliotoxin fluorocitrate, which inhibits the enzyme aconitase, in turn disrupting the tricarboxylic acid cycle and inducing metabolic arrest in astrocytes (Henneberger et al., 2010; Martín et al., 2007; Paulsen et al., 1987). This approach has previously been established to render astrocytes inactive and prevent the effects of astrocytic GAT (Boddum et al., 2016; Bonansco et al., 2011).”*

P11 *“Although the mechanisms and specificity of fluorocitrate actions are incompletely understood, and prolonged treatment with fluorocitrate has the potential to compromise neuronal integrity, this pharmacological approach remains one of the few tools available to render astrocytes and their transporters inactive. Furthermore, exposure over short durations (1 hr), as used here, is thought to have limited effects on downstream neuron viability (Henneberger et al., 2010; Martín et al., 2007).”*

4. *Some of the fonts are very small. For example, in Figure 4.*

RESPONSE: We have increased the font size where appropriate across all figures.

Overall, a very nice paper that requires some simple revisions.

Thank you.

Reviewer #3 (Remarks to the Author):

This study builds upon a previous one from the same group (Lopes et al., 2018), which revealed presynaptic modulation of dopamine (DA) release from midbrain axons in striatum by GABA using direct pharmacological manipulation of GABA_A and GABA_B receptors. Here, Roberts and colleagues now show that endogenous elevation of GABA in striatal slices following pharmacological block of the neuronal and astrocytic membrane GABA transporters GAT1 and GAT3 also depresses DA release in dorsolateral striatum (DLS) more so than in nucleus accumbens (NAc). The authors go on to show that most GAT proteins in striatum distribute to astrocytes in DLS and that pharmacological poisoning of astrocytes with fluorocitrate occludes the inhibition of DA release by the non-selective GAT blocker NPA, pointing to astrocytes as the main source of extrasynaptic GABA produced by broad pharmacological antagonism of GAT in DLS.

RESPONSE: We would like to start by bringing to the reviewer’s attention some areas that might have been subject to misplaced deductions, which in turn might have negatively coloured the review. We would like to take this opportunity to reiterate the facts as we see them, so that any potential misunderstandings can be moderated and all parties are fully informed as to the nature of our discoveries and their novelty. We have underlined two sentences from above to respond to. Firstly, we do not test, show or claim that *“most GAT proteins in striatum distribute to astrocytes.”* **Rather**, our data indicate that the GATs in striatum regulating GABA receptor-mediated inhibition of DA release are at least partly astrocytic, because the effects of GAT inhibition (with NPA) are occluded during astrocyte inhibition with fluorocitrate. We also show images of GATs on other (neuronal) structures. We hope that the revisions we have made throughout the manuscript help to clarify these points.

Secondly, we do not think that *“astrocytes are the main source of extrasynaptic GABA”*. **Rather**, astrocytes are important in determining the extrasynaptic concentration of GABA through uptake of extracellular GABA by GAT. The manuscript clarifies that there are **many sources of striatal GABA** and for example we clearly state in the Introduction that *“The striatum contains a high density of GABAergic projection neurons and interneurons and, in addition, receives a source of GABA co-*

released from mesostriatal DA neurons”. We also show that the source of GABA involved in regulating DA is GAD-dependent, consistent with a neuronal source (**Fig. 1** in revised manuscript).

They then make the interesting observation that in a mouse model of Parkinson’s disease, prior to overt neurodegeneration, DA and GABA release from midbrain dopaminergic axons is decreased in DLS as a result of excessive buildup of GABA in striatum, possibly caused by downregulation of striatal GAT1 and GAT3 protein levels. Overall, this study makes a compelling case for endogenous accumulation of GABA being capable of exerting a negative modulatory action on synaptic release of DA and GABA from midbrain dopaminergic axons, and for this mechanism contributing to diminished DA signalling in early parkinsonism. That said, many of this study’s findings repeat prior published reports or lack experimental support, making it not suitable for publication in Nature Communications.

RESPONSE: We feel that the reviewer’s claim, which we have underlined, that “*many of this study’s findings repeat prior published reports or lack experimental support*” is unjust and misplaced. We included some data in our original manuscript that corroborated some limited but key observations previously made, in order to provide a rigorous foundation and essential comparison data in our hands for the **substantial volume of original data** that we included. We detail these areas in the specific comments below. The “*lack of experimental support*” claimed is very difficult for us to understand, especially in light of such positive comments from the other two reviewers. All of our experiments were carefully performed, and all of our interpretations and conclusions are **supported fully by evidence** that we obtained, often through multiple converging strands.

MAJOR COMMENTS:

1. *Novelty: Significant sections of this manuscript repeat previous findings already published by this group and others. While this is generally a good thing, it does leave this study relatively light on original material. Specifically, the authors had already published a study (Lopes et al. 2018) showing that GABA inhibits electrically and optogenetically evoked DA release in DLS independent of local cholinergic signaling, which is now repeated in significant sections of Figure 1 and in Fig 2K.*

RESPONSE: We strongly contest the reviewer’s perception that our study is “*light on original material*”. Of 52 key sub-panels of data presented in the original figures, **41 panels contained original findings**. Where we repeated some prior findings, it was to corroborate and **establish a foundation** upon which our new experiments and observations built. For the specific figures of the original manuscript cited by the reviewer- Fig. 1 and Fig. 2K –, those panels extended our previously published study (Lopes, Roberts et al. 2019) to make entirely novel findings. In Fig. 1, only 2 of 9 data panels repeated previous observation. We corroborated that there was tonic GABA receptor-mediated inhibition of DA release in dorsal striatum as shown in Lopes, Roberts et al, to then show for the first time that: this extends to the ventral striatum; it arises from a GAD-dependent GABA source; and does not arise from a ALDH-dependent GABA source, i.e. GABA co-release from DA axons. This is a **large set of original findings**. Fig. 2K compared the effect of GABA in CPu with NAc; these are new data. Nonetheless, and in the spirit of trying to achieve an optimal balance in the paper, we have **revised figure 1 layout**, to move some of the corroborating foundational data to Supplementary Figure 1a (electrical stimulation and GABA-receptor antagonist effect in DLS). For information, none of the data shown in the revised manuscript have been published previously, and 47 out of 53 data panels in the main figures are new observations.

A logical extension of this previous work is that other means of elevating GABA receptor signalling in slice ought to have comparable effects, including block of the membrane GABA transporters GAT1 and GAT3, which have been extensively shown elsewhere to increase extracellular GABA levels in striatum (Ade et al. 2008; Cepeda et al 2013; Kirmse et al 2008; Santhakumar et al 2010). The findings in Figure 2 are therefore not surprising, and those in Figure 3 merely replicate of those previous studies.

RESPONSE: We do not view the perception of “surprise” to be of value in assessing novelty. While some aspects of our new findings are indeed a “logical extension of ...previous work,” a link between GATs and striatal DA release is fundamentally novel and, regardless of whether surprising or logical, has **never been explicitly tested** and **cannot be assumed**. Our findings in **Fig. 2** provide this evidence, and we then build further with additional new understanding.

In **Fig. 3**, only a limited number of sub-panels corroborate previous findings (as we state clearly in the text). It was imperative that we corroborate in our hands that GATs determine extrasynaptic GABA concentrations in dorsal striatum (panel a, corroborative data) in order to be able to compare the scenario in DLS with our new data in NAc (panels b,c, novel data), in order to correlate with our voltammetry and immunohistochemistry data and to be able to make an appropriate comparison between regions. An omission of our own data from DLS would have prevented the reader from gaining an appropriate understanding of the difference in GAT regulation of GABA tone between DLS and NAc.

2. Regional differences: This manuscript reveals that tonic GABA signalling in slice hampers DA release in both DLS and NAc (Figure 1), but that only DLS is susceptible to further increases in presynaptic inhibition by GABA following NPA treatment (a non-selective antagonist of GAT1 and GAT3; Figure 2). They conclude that GAT does not significantly limit extrasynaptic GABA levels in NAc, and support this claim using immunohistochemistry (Figure 4), showing a slight but significant “enrichment” in DLS relative to NAc for GAT1 and GAT3. This figure is highly deceiving, as the authors normalized GAT immunofluorescence within striatum to min/max values. While this can be used to reveal slight differences, it cannot be used to comment on the magnitude or biological relevance of the said difference.

RESPONSE: We are happy to **clarify** our analyses of the immunohistochemistry in **Fig 4**. For the purposes of illustration, we normalized the heat maps of immunoreactivity to min/max values, but these values were **not** used for the quantitative analyses. The data that indicate a significant enrichment of GATs in DLS (**Fig. 4b, Fig 4d** in revised manuscript) are normalised to **medians**, not to min/max values. Analysing to the median grey values in each hemisphere allowed us to control for differences within and between animals. We have ensured that our analyses of these data are clearly described in the Methods section. Every care was taken to ensure that parameters were kept consistent across tissue processing, imaging and analysis. The biological relevance of the significant differences ($p < 0.01$) in GAT immunoreactivity [across DLS and NAc] stems from this rigorous data acquisition and analyses.

Moreover, the histograms in panels B and D only span 0.8-1.2 a.u. to further accentuate small differences.

RESPONSE: We used this style of graph to help convey the key message that there are significant differences. We included y-axis breaks at the origin to be completely **transparent** about the data range illustrated.

Consistent with this, the authors show in Figure 3 that NPA doubles extrasynaptic GABA tone in NAc. Thus, GATs do limit GABA accumulation in NAc, and the absence of presynaptic modulation of GABA release by NPA revealed earlier (Figure 2) ought to be attributed to something other than GATs. This point is currently not made clear, and their conclusion of an effect specific to DLS is stretched, at best.

RESPONSE: The reviewer is correct that NPA doubles the GABA_A receptor-mediated tonic inhibition of SPNs in the NAc. The effect in DLS is four-fold, and statistically stronger than the effect in NAc ($p < 0.01$, Fig. 3). But importantly, the primary focus of our manuscript is **not** how GATs govern tonic inhibition of SPNs (via GABA_A receptors), but rather, how GATs impact on tonic inhibition of **DA**

release (involving both GABA_A and GABA_B receptors), which will involve different receptors, sites and mechanisms. Importantly, our electrophysiological, voltammetric and anatomical data are consistent with **greater GAT control** of extrasynaptic striatal GABA tone regulating DA release in the dorsal vs ventral striatum. We confirmed that NPA exerts its effects on DA (and SPNs) through increasing GABA receptor tone (**Fig. 2** and **Fig. 3** in revised manuscript). A potential difference between impact of GATs on DA release versus SPNs could potentially be explained by many mechanisms beyond the scope of the current study, such as different effects on different cell types (SPNs and DA axons) mediated by different GABA receptors and mechanisms.

Importantly, previous studies reported differences in the ability of direct and indirect pathway MSNs to detect extrasynaptic/tonic GABA (Ade et al 2008; Santhakumar et al 2010); because this study does not distinguish direct and indirect pathway MSNs and because their sample data is exceedingly small (N=6), the small difference in NAc and DLS may reflect a sampling bias rather than true areal differences. In addition, because extracellular GABA levels are not measured directly (but rather via a standing GABAergic conductance in MSNs), the effect reported in Figure 3C may not reflect differences in extracellular GABA as much as differences in the sensitivity of MSNs of both pathways to extrasynaptic/tonic GABA in DLS vs. NAc. Together, these concerns mitigate the significance of the major point made in Figures 1 through 4.

RESPONSE: It has indeed previously been shown that there are differences in the tonic GABA_A receptor-mediated conductances observed in direct and indirect pathway SPNs, arising from a resting ambient GABA tone when GATs are intact (i.e. Ade et al 2008, Jansen et al 2009). However, we can **reassure the reviewer** that Tritsch et al. (2014) showed that there is **no difference** in the effect of GAT inhibition on tonic GABA_A receptor-mediated conductance in indirect versus direct pathway SPNs (see Tritsch et al Figure 2, Supplement 2). Given these published observations, a possible sampling bias for either pathway would not underlie the difference we observed between regions (DLS vs. NAc). It is also important to reassure the reviewer here that our electrophysiological data are included to support the conclusions that can be made from independent lines of experiments with voltammetric and anatomical approaches. All of these different approaches converge on the same conclusion that there is a greater GAT control of extrasynaptic striatal GABA tone in the dorsal striatum vs ventral striatum. This conclusion does not rest on a possible differential impact of GATs on direct versus indirect pathway SPNs.

3. Assuming GAT effects to striatal astrocytes: Immunofluorescence is used to localize GAT protein to astrocytes, but the evidence provided is underwhelming. First, existing gene and protein expression data already make a compelling case for it. Second, in Figure 5A-B, the authors conclude that antibodies to S100β can be successfully used to label striatal astrocytes, but no counterstain is provided to support their argument. What if some fraction of labeled somatas were neurons?

RESPONSE: There is indeed existing transcriptomic data in support of GAT1 and GAT3 expression by astrocytes and we have ensured that this information is clear in our manuscript. However, evidence in direct support of both GAT-3 and GAT-1 **protein expression** in striatal astrocytes has been lacking. We established that GAT-immunoreactivity could be localised to S100β-immunoreactive structures. We can **reassure the reviewer** that S100β is a robust and highly-selective marker for astrocytes in striatum and elsewhere in the brain (Tong et al., 2014; Srinivasan et al., 2016; Chai et al., 2017; Octeau et al., 2018; Yu et al., 2018; Khakh, 2019; Michetti et al., 2019; Nagai et al., 2019) and we cite Chai et al (2017) as a definitive study identifying that **S100β is an astrocytic marker in striatum**. S100β has been shown to co-localise with other markers of astrocytes but not the neuronal marker NeuN (Srinivasan et al., 2016; Chai et al., 2017; Nagai et al., 2019). Nonetheless, we have now **revised our manuscript** and **included our own data** using a counterstain with the neuronal marker NeuN to corroborate the lack of detectable S100β in striatal neurons (**new Fig. 5b**). This panel also makes clearer the distinctive small size and structure of the S100β-immunoreactive cells in striatum.

This body of data argue strongly against S100 β -immunoreactive structures being neurons and should address the Reviewer's concern.

Third, their co-stain with antibodies to GAT1 and GAT3 In Fig 5C-D is amongst the least convincing I have seen. GAT fluorescence is not seen within S100B+ cell bodies, and instead looks to be distributed in fine processes that are adjacent to S100B+ processes, with little to no right overlap. This is more consistent with astrocytic processes enveloping GAT+ axonal varicosities.

RESPONSE: We thank the reviewer for this valuable feedback, which was likely compounded by the poor image resolution on the submitted compressed image, and we are glad to have the opportunity to address this shortcoming, in several ways. It has previously been shown that GAT-3 immunoreactivity is more prominent on the surface of astrocytes than within their cell bodies (Yu et al., 2018, see their Figure 5). Intuitively, it would be advantageous for cells to traffic GATs to their plasma membranes in order for the GATs to fulfil their primary role of removing GABA from the extracellular space. In the case of the S100 β -expressing striatal astrocytes we studied, our imaging data are certainly consistent with an enriched expression of GAT-1/-3 immunoreactivities on the plasma membranes of these cells. However, to better assuage the reviewer's concerns we have now substantially **revised our figures and text** and included: additional single-plane confocal images of three different examples of S100 β + astrocytes, for both GAT-1 and GAT-3 immunolabelling experiments, which together provide clear and compelling evidence of plasma membrane expression of GATs (**Fig. 5**); as well as images of these S100 β + astrocytes in multiple z-planes (**Supplementary Fig. 8**) in order to better demonstrate that GAT-1 and GAT-3 expression is uniform in width and distributed across large areas of the plasma membrane of each cell. Our revised text now states on P7 that GAT-3 "GAT immunoreactivity was observed distributed over a large surface area of plasma membranes when assessed in three dimensions". We emphasise that the uniform, contiguous labelling of GATs over large surface areas of S100 β -expressing astrocytes that we show here is not consistent with individual GAT+ neuronal fibres running a tight course along an astrocyte plasma membrane. We hope also that the resolution of these uploaded images is now sufficient to appreciate these findings.

Fourth, no quantification is provided.

RESPONSE: Quantification was not a focus of this aspect of the study. The expression of GAT1 and GAT3 on the plasma membranes of astrocytes is consistent with the literature and with the functional roles that we attribute to these GATs and to astrocytes in our paper.

Thus, the only evidence pointing to astrocytes as being the main target of NPA and contributors of tonic GABA is the fluorocytrate experiment, whose effects on neurons cannot be ruled out. The authors should therefore refrain from constantly referring to extracellular GABA as being exclusively astrocyte-derived, they should rephrase their title and update their discussion to acknowledge this point.

RESPONSE: We flag here a misinterpretation of our data and the conclusions we drew from them. As we state above, we do not suggest or imply that GABA is derived from astrocytes, and we describe clearly the many neuronal sources of GABA. We have checked that our manuscript does not give this false impression. Rather, our voltammetric, pharmacological and anatomical data converge to make a compelling case for the functional importance of **GATs** localised on astrocytes.

4. Absence of GAT molecules in dopamine axons: Although peripheral to their main story, the authors make a point to challenge a previous finding that co-release of GABA from DA axons requires GAT-mediated presynaptic uptake of GABA (Tritsch et al 2014). This earlier study used GAT blockers and reported a near complete loss of GABA co-release. The present study finds that GAT blockade depresses DA release by 30-40%

via presynaptic GABA receptors, indicating that some presynaptic inhibition may have contributed to the co-release phenotype. However, the Tritsch et al study controlled for such confound by using other means of elevating GABA tone and by limiting presynaptic modulation with GABA_B receptor antagonists.

RESPONSE: We thank the reviewer for this input. We can reconcile these different findings. While Tritsch et al. (2014) do control for the confounding effects of GABA_B receptor activation in decreasing GABA co-release from DA axons following GAT blockade, they do not control for the additional effects of GABA_A receptors, as they needed to record a GABA_A receptor-dependent current in SPNs as a measure of co-release. We have recently shown that both GABA_A and GABA_B receptors decrease transmitter release from DA terminals (Lopes, Roberts et al., 2019) and therefore it is likely that Tritsch et al could not fully control for effects of GAT blockade on release from DA axons.

Nevertheless, Roberts and colleagues suggest that DA axons do not express GAT (challenging the mechanism championed by Tritsch et al) using immunohistochemistry, which is inherently limited in its ability to categorically rule out protein expression, as the technique can only provide relative comparisons given experimental variables like antibody sensitivity and a microscope's imaging settings.

RESPONSE: We recognise that lack of proof for GAT protein expression on DA axons is not proof of absence, and in interpreting our immunohistochemical data (P15) we state only that “*We did not find evidence for robust localization of GAT-1 or GAT-3 to DA axons*”. This lack of evidence together with our collective other main findings provide an alternative potential explanation for the interpretation provided in Tritsch et al. (2014) that GATs mediate GABA uptake for GABA co-release. GAT involvement in GABA co-release from DA axons could occur via the regulation of GABA tone acting to inhibit axonal release. This insight is supplemental to our main findings, and the novelty and impact of our manuscript does not depend on it, but we think that we simply owe it to the field to communicate this alternative potential explanation. We have **checked that our description in the revised manuscript** is presented in a suitably accurate, reserved and scholarly way that does not overblow our suggestion. In addition, we have provided **additional new figures** at higher power for assessing co-localisation of GAT-1 and GAT-3 to dopamine axons for which we still do not find evidence (**Supplementary Fig. 6** in revised manuscript). We controlled for the variables mentioned (antibody sensitivity, imaging settings) by using the appropriate controls and keeping imaging parameters consistent (details in Methods). Furthermore, using these same variables (antibodies and imaging settings) we do find and report robust evidence of expression of GAT1 protein on the axons of PV-interneurons in the striatum, which we employed as a positive control (**Supplementary Fig. 7** in revised manuscript). Also, it is worth noting that the variables important for immunofluorescence (sensitivity of ‘probes’, imaging settings) are also important for fluorescent in-situ hybridisation (FISH), the technique used in Tritsch et al., (2014). In our view, however, FISH is one step further away from mechanism, which must eventually be actioned by proteins.

If the authors insist on challenging the results of the Tritsch et al. (2014) study, they ought to do so directly, perhaps by repeating their experiments in GABA_B antagonists and showing that any means of increasing extracellular GABA in striatum (like fluorocytate treatment) are sufficient to abolish GABA co-release from DA axons. If on the other hand the authors are not willing to do so directly and thoroughly, they ought to keep it to a discussion point.

RESPONSE: We hope that our cautious and careful wording of this data and issue and new data as described above now reassure the reviewer that our perceived “*challenge*” is a scholarly addition to the field, a complementary data set and alternative possible explanation for further evaluation by others in the future.

5. Figure 6 describes a set of interesting observations in a mouse model of Parkinson's disease that superficially support their model, but several key experiments well within the reach of the experimenters are missing to more conclusively connect the dots.

RESPONSE: The observations we described in this model of PD were designed to highlight the potential relevance of our findings to disease. This range of data provide hugely exciting observations that extend our original findings beyond a significant and already thorough study of the intact striatum, to bridge the gap to disease relevance. The reviewer has requested a set of data and experiments that while being “within our reach” could support a further and independent manuscript and are not entirely necessary for the integrity of our findings as presented. The data we have included present an exciting opening to a new line of enquiry. They are not intended to be the end of the story. We have detailed our responses to each question below and have included some **additional data**.

First, the authors do not exclude the possibility that neuronal loss contributed to the observed decrease in DA and GABA release from SNc axons. They merely cite previous work, but the experimental conditions here are different, as the authors bred this mutation to Slc6a3-Cre mice.

RESPONSE: To assess the integrity of DA neuron innervation we assessed striatal DA content in this mouse and found normal DA levels (**Fig. 7b** in revised manuscript). Since striatal DA content is maintained, but DA release is attenuated, this indicates a local striatal change in the releasability of DA that is not reflected in the capacity to produce DA. This discrepancy between striatal content and release is a critical feature of the SNCA-OVX mouse paralleled in our crossed mouse used here, and is the phenotype we were seeking for the questions we addressed. There is no expectation that one copy of a targeted Slc6a3-IRES-Cre allele would be detrimental to neuron survival and thus, we did not perform dopamine cell counts. This should not in any way undermine the validity of the striatal observations.

Second, the authors report decreased GAT expression in striatum by western blot, but this decrease cannot be ascribed to astrocytes, as their conclusions consistently suggest.

RESPONSE: As the Reviewer indicated above in point 3, it is already appreciated that GATs can be localised to astrocytes and that GAT-3 protein in striatum is primarily localised to striatal astrocytes (Chai et al., 2017; Ng et al., 2000; Yu et al., 2018). It is therefore extremely likely that the loss of striatal GAT-3 by Western blot is a reflection of decreased GAT-3 expression in the astrocyte population. We do however readily acknowledge that the decrease in GAT-1 cannot conclusively be ascribed to a decrease GAT-1 on astrocytes. We have **checked our revised manuscript** that we do not claim that the observed decrease in striatal GATs is entirely astrocytic. Rather, we are careful to describe how our data strongly support a **major role for astrocytic GATs**, but this conclusion does not by itself exclude GATs expressed elsewhere. We point the reviewer also to **additional experiments and new data (Fig. 8i-k)** in our parkinsonian model that we think further emphasises a major role for astrocytic GATs. Please see response to next point for description.

Third, the authors do not directly demonstrate that extracellular levels of GABA are actually elevated in SNCA+ mice. Recordings similar to those shown in figure 3 are warranted. Fourth, does fluorocitrate occlude the effect of GABAR antagonists on DA shown in Figure 6H?

RESPONSE: We did not compare GABA holding currents in SPNs in SNCA+ versus control mice, as the focus of our study is on the mechanisms underlying DA release. Instead, in response to this reviewer comment, we **conducted additional experiments** to test whether by impairing astrocyte function with fluorocitrate we could equalise the effect of GABA receptor antagonists on DA release in

control and parkinsonian mice. Indeed, in a **new figure panels (Fig. 8i-k in revised manuscript)** we show that fluorocitrate boosts tonic inhibition strongly in *snca*^{-/-} but not *SNCA*⁺ mice and abolishes the differential effects of GABA receptor antagonism between genotypes. These data help to more directly implicate astrocytic GATs in contributing to the increased GABAergic inhibition of DA release and should reassure the reviewer further. They are also a valuable addition to the manuscript that further support astrocytic dysregulation in parkinsonism.

Fifth, is GABA co-release in Figure 6E also subject to presynaptic inhibition at baseline. In other words, does application of a GABA_B receptor antagonist potentiate GABA co-release? If so, does applying a GABA_B receptor antagonist to SNCA⁺ mice restore IPSCs to the same amplitude as controls? If not, it would be highly suggestive of additional mechanisms taking place, which ought to be acknowledged.

RESPONSE: While understanding whether GABA co-release in *SNCA*⁺ can be restored by a GABA_B receptor antagonist would be an interesting follow-on study, it will not allow assessment of the likely full range of presynaptic GABA receptors that control GABA co-release: both GABA_A and GABA_B receptors regulate DA release. Since GABA_A antagonists will obscure the IPSC, we cannot readily assess whether the antagonists for both receptors restore IPSCs to make sufficiently useful insights, and we therefore think that studying GABA_B effects alone this experiment goes beyond the scope of this manuscript.

MINOR COMMENTS

1. The authors seem to assume that ALDH1a1 is only expressed in DA neurons and that disulfiram selectively blocks GABA synthesis in these cells. However, ALDH1a1 is abundantly expressed throughout the brain by astroglial cells (<https://www.proteinatlas.org/ENSG00000165092-ALDH1A1/tissue/mouse+brain>), and ALDH1a1's function is not limited to producing GABA. Moreover, disulfiram is far from a selective ALDH1a1 antagonist. This manipulation will therefore impact many dehydrogenases throughout the brain, possibly confounding the author's interpretation of the data.

RESPONSE: We thank the reviewer for this comment but can reassure that we are aware that ALDH1a1 is expressed elsewhere e.g. by astrocytes (Chai et al., 2017; Zhang et al., 2014) and that disulfiram is not an ALDH1a1-selective antagonist. As is a caveat with all pharmacology experiments, disulfiram might have off-target effects. However, since disulfiram does limit GABA co-release from DA axons, we could use it to test one discrete question: does a reduction in GABA co-release from DA axons reduce the GABAergic inhibition of DA release. In disulfiram-treated slices, we validated that GABA co-release was reduced by 50% (**Supplemental Fig. 2**; see also Kim et al., 2015), and despite this established effect, we did not find a corresponding reduction in the effect of GABA receptor antagonists on DA release (**Fig. 1f** in revised manuscript), indicating that GABA co-release, **or any ALDH-dependent source of GABA for that matter**, is not required for tonic inhibition of DA release. Furthermore, we performed the complementary experiment to limit canonical GAD-dependent sources of GABA, and by contrast inhibition of GAD was paralleled by a reduction in tonic inhibition of DA indicating a role for GAD-dependent sources (Fig. 1g,h). To help further alleviate the reviewer's concern we have **also modified the text** to clarify that disulfiram is "*a non-specific ALDH inhibitor*". Also, in further support of the hypothesis that tonic inhibition of DA does not require GABA co-release is our data showing that in *SNCA*-OVX mice, GABA co-release from DA axons is attenuated (**Fig. 7e-f** in revised manuscript) but tonic inhibition of DA release is increased (**Fig. 8a** in revised manuscript).

2. In Fig. 1E, the authors rely on baseline-normalized data to show that GABA receptor antagonists have a greater effect on DA release in the presence of disulfiram compared to control. It is however possible that one of disulfiram's many effects is to elevate tonic GABA levels, and thus depress DA release more strongly at

baseline. The authors ought to present and statistically compare raw values before bath application of GABAR antagonists to exclude this possibility. The authors also ought to repeat the measurements of tonic GABA in MSNs (as in Figure 3).

RESPONSE: We thank the reviewer for giving us the chance to include the extra control data. We have now **included additional data** to show that disulfiram alone did not significantly change underlying evoked [DA]_o levels (new **Supplementary Fig. 2c**).

3. For experiment comparing before and after drug application, the authors ought to use a Wilcoxon signed rank test (not Mann-Whitney). Such comparison may reveal a statistically significant effect of NPA on fluorocitrate-treated slices in Figure 5G.

RESPONSE: Thanks for this suggestion, but the data shown in Figure 5g (now **Fig. 6c** in revised manuscript) are not obtained using a before-and-after design in given sites, but rather, are obtained in different slices pre-incubated in either fluorocitrate or vehicle, then subsequently with or without NPA, as explained for this set of experiments in the methods. Because of the way these data were collected, we ran an unpaired analysis for data which were not normally distributed, i.e. a Mann-Whitney test. For other experiments in the manuscript where we compare drug effect to drug-free time-matched controls, we have modified some of our statistical analyses in the revised manuscript to compare drug and drug-free controls using a two-way repeated measures ANOVA with, where appropriate, multiple comparisons with Sidak's correction.

References cited in response:

- Ade, K.K., Janssen, M.J., Ortinski, P.I., and Vicini, S. (2008). Differential Tonic GABA Conductances in Striatal Medium Spiny Neurons. *J. Neurosci.* 28, 1185–1197.
- Boddum, K., Jensen, T.P., Magloire, V., Kristiansen, U., Rusakov, D.A., Pavlov, I., and Walker, M.C. (2016). Astrocytic GABA transporter activity modulates excitatory neurotransmission. *Nat. Commun.* 7, 13572.
- Bonansco, C., Couve, A., Perea, G., Ferradas, C.A., Roncagliolo, M., and Fuenzalida, M. (2011). Glutamate released spontaneously from astrocytes sets the threshold for synaptic plasticity. *Eur. J. Neurosci.* 33, 1483–1492.
- Chai H, Diaz-Castro B, Shigetomi E, Monte E, Oceau JC, Yu X, Cohn W, Rajendran PS, Vondriska TM, Whitelegge JP, Coppola G, Khakh BS (2017) Neural Circuit-Specialized Astrocytes: Transcriptomic, Proteomic, Morphological, and Functional Evidence. *Neuron* 95:531-549.e9.
- Chazalon, M., Paredes-Rodriguez, E., Morin, S., Martinez, A., Cristóvão-Ferreira, S., Vaz, S., Sebastiao, A., Panatier, A., Boué-Grabot, E., Miguez, C., et al. (2018). GAT-3 Dysfunction Generates Tonic Inhibition in External Globus Pallidus Neurons in Parkinsonian Rodents. *Cell Rep.* 23, 1678–1690.
- Henneberger, C., Papouin, T., Oliet, S.H.R., and Rusakov, D.A. (2010). Long-term potentiation depends on release of d-serine from astrocytes. *Nature* 463, 232–236.
- Khakh BS (2019) Astrocyte–Neuron Interactions in the Striatum: Insights on Identity, Form, and Function. *Trends Neurosci* 42:617–630.
- Kim, J.I., Ganesan, S., Luo, S.X., Wu, Y.W., Park, E., Huang, E.J., Chen, L., and Ding, J.B. (2015). Aldehyde dehydrogenase 1a1 mediates a GABA synthesis pathway in midbrain dopaminergic neurons. *Science* (80). 350, 102–106.
- Lopes, E.F., Roberts, B.M., Siddorn, R.E., Clements, M.A., and Cragg, S.J. (2019). Inhibition of nigrostriatal

- dopamine release by striatal GABAA and GABAB receptors. *J. Neurosci.* 2028–18.
- Martín, E.D., Fernández, M., Perea, G., Pascual, O., Haydon, P.G., Araque, A., and Ceña, V. (2007). Adenosine released by astrocytes contributes to hypoxia-induced modulation of synaptic transmission. *Glia* 55, 36–45.
- Michetti F, D'Ambrosi N, Toesca A, Puglisi MA, Serrano A, Marchese E, Corvino V, Geloso MC (2019) The S100B story: from biomarker to active factor in neural injury. *J Neurochem* 148:168–187.
- Nagai J, Rajbhandari AK, Gangwani MR, Hachisuka A, Coppola G, Masmanidis SC, Fanselow MS, Khakh BS (2019) Hyperactivity with Disrupted Attention by Activation of an Astrocyte Synaptogenic Cue. *Cell* 177:1280-1292.e20.
- Octeau JC, Chai H, Jiang R, Bonanno SL, Martin KC, Khakh BS (2018) An Optical Neuron-Astrocyte Proximity Assay at Synaptic Distance Scales. *Neuron* 98:49-66.e9.
- Srinivasan R, Lu T-Y, Chai H, Xu J, Huang BS, Golshani P, Coppola G, Khakh BS (2016) New Transgenic Mouse Lines for Selectively Targeting Astrocytes and Studying Calcium Signals in Astrocyte Processes In Situ and In Vivo. *Neuron* 92:1181–1195.
- Tong X, Ao Y, Faas GC, Nwaobi SE, Xu J, Haustein MD, Anderson MA, Mody I, Olsen ML, Sofroniew MV, Khakh BS (2014) Astrocyte Kir4.1 ion channel deficits contribute to neuronal dysfunction in Huntington's disease model mice. *Nat Neurosci* 17:694–703.
- Tritsch, N.X., Oh, W.J., Gu, C., and Sabatini, B.L. (2014). Midbrain dopamine neurons sustain inhibitory transmission using plasma membrane uptake of GABA, not synthesis. *Elife* 3, e01936.
- Yu X, Taylor AMW, Nagai J, Golshani P, Evans CJ, Coppola G, Khakh BS (2018) Reducing Astrocyte Calcium Signaling In Vivo Alters Striatal Microcircuits and Causes Repetitive Behavior. *Neuron* 99:1170-1187.e9.
- Zhang, Y., Chen, K., Sloan, S.A., Bennett, M.L., Scholze, A.R., O'Keefe, S., Phatnani, H.P., Guarnieri, P., Caneda, C., Ruderisch, N., et al. (2014). An RNA-Sequencing Transcriptome and Splicing Database of Glia, Neurons, and Vascular Cells of the Cerebral Cortex. *J. Neurosci.* 34, 11929–11947.

Reviewers' Comments:

Reviewer #2:

Remarks to the Author:

I was pleased to read the revised paper and find that they have addressed all my minor suggestions and comments from round 1. I still think this is a really nice paper and worthy of publication as is. I am sure it will be well cited and is thorough in nature. I also took some time to read the comments from reviewers 1 and 3, and find the authors have responded to them thoughtfully. There is only so much one can expect authors to do in a revision, and frequently one must assess a manuscript on the relevance and importance of its overall message. Based on these criteria, I think the paper is ready to be published. It is very nice to see a hard core basal ganglia lab include astrocytes in their considerations of relevant mechanisms. Congrats to the authors.

Reviewer #3:

Remarks to the Author:

I thank the authors for addressing my concerns and apologize if some of them resulted from misplaced deductions. The resubmitted manuscript is indeed much improved, including additional experiments needed to evaluate the proposed mechanisms and revisions to the text and figures that more accurately and effectively describe the data. In particular, the new immunofluorescence displays are much clearer and convincingly demonstrate that GAT1 and GAT3 distribute to astrocytes. However, some of my original reservations remain and my deductions, misplaced as they may be, stem in part from what I perceive to be mixed messages and key missing experiments. Allow me to stress the major ones:

My original review took issue with the fact that a role for astrocytic GATs, as opposed to neuronal GATs is not clearly demonstrated. I acknowledge this to be difficult, and therefore did not suggest experiments to distinguish these possibilities. Rather, I recommended the title and abstract be edited to better reflect the data and the authors' more measured conclusions throughout the result section. The title of this study remains unchanged to "Astrocytic striatal GABA transporter activity governs dopamine release...". The abstract then states that "Ambient GABA tone can be governed by GATs on astrocytes", asks "whether striatal astrocytes determine DA output" and concludes "important roles for GATs and astrocytes in determining DA release in striatum". There is no mention of a possible neuronal contribution, except for a vague qualifier that "GATs are partially localized to astrocytes". This may confuse readers and lead them to form misplaced deductions. If the facts, as the authors see them, are that GATs that "regulate GABA receptor-mediated inhibition of DA release are at least partially astrocytic" (authors' rebuttal) and that "astrocytic GATs support GABA uptake and [...] indirectly support DA release" (p.8 of revised manuscript), then the authors ought to revise the title and abstract to better reflect this.

In addition, I took issue with language suggesting that astrocytic GATs actively control DA release. While this may well be the case, it is not directly shown. The verb in the title remains "govern", which implies a direct, active role in controlling DA release. This study shows that poisoning GABA transporters elevates GABA, and that GABA acting on presynaptic GABA receptors modulates DA release. But this is not the same as showing that astrocytes actively control DA release through modulation of GAT function. For instance, pharmacological blockers of glycolysis will compromise neuronal action potential firing, yet it would be inappropriate to conclude that glycolysis governs neuronal discharge. I maintain that, in light of the findings of Lopes et al (2019) by the same group, any manipulation of extracellular GABA ought to have an effect on presynaptic DA release, and maintain that novelty ought to be measured by more than explicitly testing whether other pharmacological manipulations of extracellular GABA, whether physiological or not, indeed depress presynaptic DA release, as suspected.

While sensitive to the difficulties of carrying out experiments during this highly disruptive covid

outbreak, I remain perplexed with the author's refusal to consider one simple experiment that would go a long way in supporting their claim (point 5), referring to it as "not entirely necessary" and possibly forming the basis for a "further and independent manuscript". The authors subscribe to what may be described as confirmation bias regarding what experiments need or need not be done, refraining from carrying out experiments that may challenge the study's narrative: Astrocytes express GATs, GATs control tonic extracellular GABA levels, and tonic GABA inhibits DA release through presynaptic GABA_A and GABA_B receptors. The authors present multiple lines of evidence supporting this narrative throughout the paper, but on several occasions fall short of testing intermediate steps needed to rule out alternative mechanisms. For instance, the authors demonstrate convincingly in Figures 1 and 3 that there isn't a one-to-one relationship between the extent of presynaptic inhibition by GABA receptors on DA terminals, regulation of extracellular GABA by GATs and GABAergic tone in DLS and NAcC. The authors also show that, although disulfiram does not increase presynaptic inhibition of light-evoked DA release (Supplementary Fig. 2b), bath application of a GABA receptor antagonists potentiates DA release in disulfiram-treated slices more so than in control slices (Fig. 1f), suggesting the existence of other mechanisms than the one championed by the authors. This complicates interpretation of later findings in Fig. 8, where it is concluded that SNCA overexpressing mice have diminished DA release because of an exaggerated GABA tone without actually measuring it. Lastly, in Fig. 6, the authors show that fluorocitrate treatment fully occludes the effect of NPA, a non-specific GAT antagonist. This either suggests that astrocytes account for all presynaptic GABA uptake in DLS, which in their rebuttal the authors do not believe to be the case, or that fluorocitrate is not specific for astrocytes, a possibility the authors acknowledge, but not seriously enough to modify the title or abstract.

Minor comments:

- The text states on several occasions that GAT1 expression in DA axons has only been presumed based on mRNA expression but not confirmed using protein-based approaches such as immunofluorescence (p.7; 11; rebuttal). I wish to bring to light one reference that does show co-localization of the axonal DA transporter DAT and GAT1 protein in the striatum of mice (Uchigashima...Watanabe PNAS 2016).
- I had not realized until now that comparisons in Figs. 7 and 8 are between 2 mutant mice: one in which alpha-synuclein is knocked out, and another in which alpha-synuclein is overexpressed. The text treats the knockout as controls, and assigns phenotypes to the overexpressor, but it is not immediately clear why this is warranted, as both models have been shown to exhibit altered DA release. Indeed, in this study, electrically-evoked DA release is significantly elevated in snca^{-/-} and SNCA⁺ mice compared to Dat-Cre controls (Figs. 6a, 7a and S2b). Moreover, control mice appear to vary significantly in the enhancement of DA release observed upon bath application of GABA receptor antagonists, with some control groups showing effects similar to snca^{-/-} mice (compare Fig. 1c and S1a to Fig. 8a) and others similar to SNCA⁺ mice (compare Fig. S1d and S1e to Fig. 8a), making it difficult to assess which model, if any, behaves as control, and in which light the effects of NPA and FC ought to be interpreted. How do the authors explain such variability between experimental groups, but not within groups? The authors should disclose more side-by-side comparisons of other metrics in dat-cre, snca^{-/-} and SNCA⁺ mice to determine how to best interpret the data.
- Please check the figure legend of Fig 1c and Fig 1f/g: it describes different Ns despite the mean traces looking identical. If controls are drawn from the exact same experimental group, the authors should state so clearly in the legend and refrain from using different representative DA transient traces. The same applies to Figs 2a,d,e,g,h, all of which include the same exact control data set, yet the representative traces change. Note that the appropriate control groups for Fig. 2h should be slices maintained in DHBE+SKF, not slices maintained in DHBE only.

Response to Reviewers - June 2020

Many thanks to the reviewers for again taking the time to give us their thoughtful feedback.

Reviewers comments are in black italics. Our responses are in blue.

Reviewer #2

I was pleased to read the revised paper and find that they have addressed all my minor suggestions and comments from round 1. I still think this is a really nice paper and worthy of publication as is. I am sure it will be well cited and is thorough in nature. I also took some time to read the comments from reviewers 1 and 3, and find the authors have responded to them thoughtfully. There is only so much one can expect authors to do in a revision, and frequently one must assess a manuscript on the relevance and importance of its overall message. Based on these criteria, I think the paper is ready to be published. It is very nice to see a hard core basal ganglia lab include astrocytes in their considerations of relevant mechanisms. Congrats to the authors.

RESPONSE: We thank the reviewer for their enthusiastic and supportive remarks, for their previous constructive feedback which helped improve the quality of the manuscript, and for their positive assessment of our responses to other reviewers' comments.

Reviewer #3

I thank the authors for addressing my concerns and apologize if some of them resulted from misplaced deductions. The resubmitted manuscript is indeed much improved, including additional experiments needed to evaluate the proposed mechanisms and revisions to the text and figures that more accurately and effectively describe the data. In particular, the new immunofluorescence displays are much clearer and convincingly demonstrate that GAT1 and GAT3 distribute to astrocytes.

RESPONSE: We thank the reviewer for their detailed feedback and support for our revisions. We agree that our revised manuscript is improved.

However, some of my original reservations remain and my deductions, misplaced as they may be, stem in part from what I perceive to be mixed messages and key missing experiments. Allow me to stress the major ones:

My original review took issue with the fact that a role for astrocytic GATs, as opposed to neuronal GATs is not clearly demonstrated. I acknowledge this to be difficult, and therefore did not suggest experiments to distinguish these possibilities. Rather, I recommended the title and abstract be edited to better reflect the data and the authors' more measured conclusions throughout the result section. The title of this study remains unchanged to "Astrocytic striatal GABA transporter activity governs dopamine release...". The abstract then states that "Ambient GABA tone can be governed by GATs on astrocytes", asks "whether striatal astrocytes determine DA output" and concludes "important roles for GATs and astrocytes in determining DA release in striatum". There is no mention of a possible neuronal contribution, except for a vague qualifier that "GATs are partially localized to astrocytes". This may confuse readers and lead them to form misplaced deductions. If the facts, as the authors see them, are that GATs that "regulate GABA receptor-mediated inhibition of DA release are at least partially astrocytic" (authors' rebuttal) and that "astrocytic GATs support GABA uptake and [...] indirectly support DA release" (p.8 of revised manuscript), then the authors ought to revise the title and abstract to better reflect this.

RESPONSE: We thank the reviewer for clarifying some of their perceptions of mixed messages that might confuse other readers. We are of course keen to avoid any confusion, and so in the light of these comments we have **revisited our wording to ensure that our main messages are clearer.**

*Firstly, our manuscript has always set out to show that GATs on astrocytes are strongly involved in supporting DA release, but has never excluded additional roles for GATs on neurons. We have always included descriptions of known GAT localisations in striatum, to astrocytes and neurons, as well as our own images supporting GAT-1 localisation on PV+ neurons (Fig. S7) as well as on astrocytes. We have never made the claim that GATs located *exclusively* to striatal astrocytes support DA release but rather we have tested whether GATs on astrocytes are important, which need not be at the exclusion of other, neuronal sites of GAT localisation, which we did not test. We provide evidence that GAT-mediated support*

of striatal DA release appears to be mediated by GATs located at least in part on astrocytes: e.g. GAT-3 regulates DA and is located on astrocytes (Figs. 2, 5); the effects of GAT inhibitors are occluded in the presence of astrocyte inhibitor fluorocitrate (Fig. 6a-d); and astrocyte inhibition augments ambient GABA inhibition of DA release (Fig. 8i-k). We are keen to ensure that our main message is not misinterpreted by a few words in the opening sentences, and therefore, because of the Reviewer's ongoing concern about this we have as requested now **modified the title** to move the wording "astrocytic" from the title, and instead we first introduce the importance of astrocytes in the abstract where there is scope to clarify our key message without misinterpretation. In the **abstract** we have also explicitly added "*and neurons*" and made other small edits to compensate for word count. In the discussion p12, we have **added an additional explicit statement** about the potential roles of GATs on neurons that "*our analyses do not exclude similar roles for GATs located on neurons.*" We hope these changes now clarify our intended messages.

In addition, I took issue with language suggesting that astrocytic GATs actively control DA release. While this may well be the case, it is not directly shown. The verb in the title remains "govern", which implies a direct, active role in controlling DA release. This study shows that poisoning GABA transporters elevates GABA, and that GABA acting on presynaptic GABA receptors modulates DA release. But this is not the same as showing that astrocytes actively control DA release through modulation of GAT function. For instance, pharmacological blockers of glycolysis will compromise neuronal action potential firing, yet it would be inappropriate to conclude that glycolysis governs neuronal discharge.

RESPONSE: We thank the reviewer for clarifying that our wording "govern" might be misconstrued as showing that astrocytes actively and dynamically dictating DA release in some leading capacity. We did not intend for dictation to be our message; rather, as the rest of our manuscript puts into context, we show for the first time that astrocytes play a strongly *supportive* role that regulates DA output. Therefore, we have **standardised our language by replacing** the handful of instances of the word "govern" with words such as "support" or "regulate" that we had used elsewhere.

I maintain that, in light of the findings of Lopes et al (2019) by the same group, any manipulation of extracellular GABA ought to have an effect on presynaptic DA release, and maintain that novelty ought to be measured by more than explicitly testing whether other pharmacological manipulations of extracellular GABA, whether physiological or not, indeed depress presynaptic DA release, as suspected.

RESPONSE: We are happy to remind all of the **many such lines of novelty** that extend our findings beyond the concept that regulators of GABA can also regulate DA release. Our study significantly expands upon the first observations that striatal GATs set the level of GABA inhibition of DA release, by finding that: striatal astrocytes are involved in this mechanism, and therefore locally support DA release; and that GABA and DA co-release are reduced and tonic inhibition of DA release is augmented in a mouse model of early parkinsonism; together revealing that the regulation of striatal GABA-DA interactions by GATs and astrocytes are novel loci for dysfunction in Parkinson's disease. **These are substantial further findings that represent timely and novel advances.** We also do think that the fundamental link between GATs and striatal DA release **is novel**, even if it might be suspected, because it has never been explicitly tested and cannot be assumed.

While sensitive to the difficulties of carrying out experiments during this highly disruptive covid outbreak, I remain perplexed with the author's refusal to consider one simple experiment that would go a long way in supporting their claim (point 5), referring to it as "not entirely necessary" and possibly forming the basis for a "further and independent manuscript". The authors subscribe to what may be described as confirmation bias regarding what experiments need or need not be done, refraining from carrying out experiments that may challenge the study's narrative: Astrocytes express GATs, GATs control tonic extracellular GABA levels, and tonic GABA inhibits DA release through presynaptic GABA_A and GABA_B receptors. The authors present multiple lines of evidence supporting this narrative throughout the paper, but on several occasions fall short of testing intermediate steps needed to rule out alternative mechanisms. For instance, the authors demonstrate convincingly in Figures 1 and 3 that there isn't a one-to-one relationship between the extent of presynaptic inhibition by GABA receptors on DA terminals, regulation of extracellular GABA by GATs and GABAergic tone in DLS and NAcC. The authors also show that, although disulfiram does not increase presynaptic inhibition of light-evoked DA release (Supplementary Fig. 2b), bath application of a GABA receptor

antagonists potentiates DA release in disulfiram-treated slices more so than in control slices (Fig. 1f), suggesting the existence of other mechanisms than the one championed by the authors. This complicates interpretation of later findings in Fig. 8, where it is concluded that SNCA overexpressing mice have diminished DA release because of an exaggerated GABA tone without actually measuring it.

RESPONSE: Thanks for flagging these specific points. We can unpack and address each example given.

Firstly, the reviewer's original point 5 was several suggestions for additional experiments concerning the role for astrocytes and/or GABA tone in our observations in a mouse model of Parkinson's disease. We did undertake additional experiments in response, to provide further support for our hypothesis that astrocytes can support DA output, and we included data as suggested, to show that differences between a mouse model of PD and controls could be abolished by impairing astrocyte function with fluorocitrate (Fig. 8i-k). The reviewer does not make it entirely clear now what the further "one simple experiment" is that we do not consider, but having deliberated over their collected comments here and in their previous review we deduce they are wanting us to "actually measure" GABA tone in SNCA+ mice by testing whether tonic GABA_A receptor-mediated currents in spiny projection neurons are modified in SNCA+ mice compared to *Snca*^{-/-} controls. We did consider this experiment, but tonic GABA_A currents on GABAergic spiny neurons only assess GABA tone at those specific receptors on those specific neurons, and would not provide a direct or definitive insight into whether GABA tone is modified at the GABA_A and GABA_B receptors on DA axons that regulate DA release. Tone on SPNs versus DA axons might easily diverge owing to different receptor types, intracellular coupling mechanisms and/or if there are distinct anatomical arrangements between sources of GABA, GATs, and target neurons. To test whether the GABA tone that controls DA release is modified requires direct recordings of DA release, as we have done. We hope the reviewer can therefore understand why we think therefore that recordings of SPNs are **not essential** for us to make conclusions relating to the GABA control of DA release. We can also reassure the reviewer about our interpretations of GABA tone in our responses to their following concerns.

The reviewer suggests that we do not test alternative hypotheses for some observations. We bring to the reviewer's attention that we explicitly rule out multiple alternative key explanations that might compete with main interpretations, that include for example whether GAT inhibitors deplete DA stores, or act at DA D₂ receptors, or via glutamate inputs, or via modulation of DA uptake, which they do not, or whether GAT inhibitors have more powerful effects in DLS than NAc due to a different responsiveness to applied GABA, which there is not. We do not test alternative mechanisms where they might be acting in addition to the mechanisms we have shown, e.g. other sites of GATs, other mechanisms that regulate GABA tone etc, because we are not seeking to identify *all* of the mechanisms that govern GABA tone. See next paragraphs for how we have **revised the manuscript** to address these concerns.

The reviewer gives examples of their perceived omissions in the steps taken or in our interpretations. They suggest firstly that we do not address an explanation for the lack of correlation between the extent of presynaptic inhibition shown by GABA-R antagonists, and the role of GATs. We presume the reviewer means between regions, and is suggesting that a greater role for GATs in DLS might be expected to tally with less tonic inhibition, which it does not obviously do. We show that GAT function correlates with regional differences in GAT levels, but we do not compare this to the extent of GABA-R antagonist effects in each region, precisely because we do not mean to assume or ever suggest that GATs will be the only mechanisms to govern GABA tone. We are not looking for all the mechanisms that govern tone, but rather we show that GATs play a role. We thank the reviewer for giving us the chance to address this point and **we have added a helpful sentence to the manuscript on p7** to clarify that "We note also that while GAT levels and function tally, they do not necessarily match the effect size of GABA-receptor antagonists in each striatal region, suggesting that additional mechanisms besides GATs regulate GABA tone and differ between regions."

The reviewer then points out that "although disulfiram does not increase presynaptic inhibition of light-evoked DA release (Supplementary Fig. 2b), bath application of a GABA receptor antagonists potentiates DA release in disulfiram-treated slices more so than in control slices (Fig. 1f) suggesting the existence of other mechanisms than the one championed". We understand this to mean that the Reviewer finds it inconsistent that on the one hand, disulfiram can slightly promote the subsequent effects of GABA

receptor antagonists (Fig. 1f), from which we suggest disulfiram has increased GABA tone, whilst on the other hand, disulfiram alone does not reduce the levels of evoked DA release (Fig. S2b), which seems to suggest that disulfiram has not increased GABA tone. We agree that at face value, such interpretations of those observations do seem in conflict, and might lead the reviewer to question whether the level of GABA tone can be indicated by the effect size of GABA-R antagonists. We thank the reviewer for the opportunity to clarify. We cannot use the effects of ALDH inhibition by disulfiram on DA release levels alone (Fig. S2b) to address whether this modifies the level of GABA tone. ALDH-inhibition using disulfiram could have multiple effects on DA release levels besides acting on GABA co-storage. It will impact on DA metabolism since ALDH participates in DA catabolism, and potentially, by reducing GABA co-storage, might have unexplored consequences for vesicular co-packing of DA. We cannot therefore make conclusions about GABA tone from the effects of disulfiram alone on evoked DA levels. Rather, we can use ALDH inhibition as a tool to reduce nigrostriatal GABA sources to address whether this impacts subsequently on the effect of GABA-R antagonists as a measure of GABA tone. Disulfiram did not reduce the effects of GABA-R antagonists, allowing us to discount GABA co-release as a primary source of GABA tone. And in fact, it slightly increased their effect, requiring us to make the speculation that GABA co-release might help to limit GABA tone. **To make this important issue clear to other readers, we have now added the following sentence, p4:** *“Disulfiram alone did not significantly modify evoked [DA]o (Supplementary Fig. 2b), but we caution that the effects of ALDH inhibition on DA release levels alone cannot be used to assess GABA tone because ALDH is also involved in DA catabolism.”*

Furthermore, in revising and clarifying these aspects of the text, and considering the necessary implications of reductions in GABA co-release leading to an increase in GABA tone, we have **also revised and clarified the corresponding sections of the discussion**, in order to refine the link we made between this finding and our observations in a PD model where there is reduced GABA co-release, enhanced tonic inhibition on DA release and reduced GAT level. We have **modified two discussion sentences** as follows. On p11, we link these findings with a resulting speculation about causality: *“The latter two findings together suggest that deficits in GABA co-release from DA neurons in our PD model might be a driving factor in leading to an enhancement of tonic GABA inhibition on DA axons by the GAD-dependent GABA tone, that in turn further compounds DA and GABA release deficits.”* On p13, where we discuss potential mechanisms that could underlie the adaptations in GAT levels in our PD model, we have streamlined our descriptions and included the potential role for changes to GABA co-release, in one all-encompassing sentence: *“It is not yet known whether these adaptations in GAT result from a potential direct interaction between α -synuclein and striatal GATs and/or astrocytes, or whether they are consequential to the reduced dopamine levels, as occurs in astrocytes in globus pallidus after profound depletion of dopamine (Chazalon et al., 2018), or to the attenuation of nigrostriatal GABA co-release we observed, which might lead to compensatory lowering of striatal GAT levels.”*

We hope these responses help to demonstrate to the Reviewer and to other readers, our interpretations, and our reasons for caution where appropriate in interpreting some data sets. We hope the Reviewer agrees that these revisions further improve the manuscript.

Lastly, in Fig. 6, the authors show that fluorocitrate treatment fully occludes the effect of NPA, a non-specific GAT antagonist. This either suggests that astrocytes account for all presynaptic GABA uptake in DLS, which in their rebuttal the authors do not believe to be the case, or that fluorocitrate is not specific for astrocytes, a possibility the authors acknowledge, but not seriously enough to modify the title or abstract.

RESPONSE: We chose to remain duly cautious with our interpretation of these data because we have not tested the role of neuronal GATs, and therefore, rather than claiming that astrocytes account for all GABA uptake, we have offered possible alternative explanations. Given that the reviewer wants us to acknowledge neuronal sites of GATs, we hope that our explanation of taking this cautious approach can now be welcomed. Our data suggest that a major component of GAT-mediated support of striatal DA release is facilitated by GATs on astrocytes, and astrocytes indeed provide the main location for GAT-3, but because we do not test the role for neuronal GATs, we do not therefore exclude their possible contribution. We hope that the requested **changes to the title and abstract (as well as our ad hoc changes to the discussion) have made this clearer** to avoid further confusion.

Minor comments:

The text states on several occasions that GAT1 expression in DA axons has only been presumed based on mRNA expression but not confirmed using protein-based approaches such as immunofluorescence (p.7; 11; rebuttal). I wish to bring to light one reference that does show co-localization of the axonal DA transporter DAT and GAT1 protein in the striatum of mice (Uchigashima...Watanabe PNAS 2016).

RESPONSE: We thank the reviewer for bringing this reference to light. The images from the referenced study suggest weak putative immunolabelling for GAT-1 in mouse DAT+ axons, although they have used locally produced GAT-1 antibodies that have apparently not been fully validated by other labs. We have however **now included this citation** in the Introduction and Discussion (**p2, p7, p12**), and have updated the accompanying Discussion paragraph (p12) with a clearer narrative and speculation about how GATs support GABA co-release.

I had not realized until now that comparisons in Figs. 7 and 8 are between 2 mutant mice: one in which alpha-synuclein is knocked out, and another in which alpha-synuclein is overexpressed. The text treats the knockout as controls, and assigns phenotypes to the overexpressor, but it is not immediately clear why this is warranted, as both models have been shown to exhibit altered DA release. Indeed, in this study, electrically-evoked DA release is significantly elevated in *snca*^{-/-} and *SNCA*⁺ mice compared to *Dat*-Cre controls (Figs. 6a, 7a and S2b).

RESPONSE: The mouse α -synuclein-null mouse is routinely used as the standard control line for the human α -synuclein overexpressor line, which is on a mouse α -synuclein-null background, in order to understand the impact of human α -synuclein (see Janezic et al. 2013, PNAS 110, 4016-4025; Dodson et al. 2016, PNAS 113, 2180-2188). We did fully describe the mouse genotype in the Results narrative (p8), Methods (p13) and figure legends. However, to help maximise clarity, we have **carefully reworded our descriptions of mouse lines** in the Methods and Results (p8-9, p13-14). We have also clarified throughout that our *SNCA*⁺ and *Snca*^{-/-} mice are also heterozygous for *Slc6a3*^{RES-Cre}. Furthermore, contrary to the reviewer's comment, the α -synuclein-null mouse is not different from wild-type mice in DA storage, DA release or behaviour (Senior et al. 2008, EFN 27, 947-957). Given the reviewer's comment, we have **added this useful background information to the mice descriptions in the Results (p8-9)**. Also, the reviewer makes the observation that DA release "is significantly elevated in *snca*^{-/-} and *SNCA*⁺ mice compared to *DAT*-Cre controls (Figs. 6a, 7a and S2b)". However, this comparison has not and cannot be made with these data for multiple reasons that we can clarify. Firstly, the experimental conditions differed between those data sets: electrically evoked release in Fig. 6a and S2b is in wild-types and is conducted in the presence of nAChR antagonist DH β E (to remove the well-established confounding effects of striatal ACh, Rice and Cragg 2004) which reduces DA release levels, whereas in Fig. 7a, recordings comparing *Snca*^{-/-} with *SNCA*⁺ mice are conducted in the absence of DH β E (consistent with previous studies in *SNCA*-OVX mice, Janezic et al 2013). Secondly, even if conditions were the same, we cannot meaningfully compare levels of DA release between experiments in different genotypes across figures, as these experiments were not conducted with the design required for meaningfully comparing DA levels with FSCV. In experiments to compare absolute DA release levels between genotypes, it is advisable to use the same electrodes in both genotypes prepared on the same day, recorded side-by-side in the same recording rigs, and sample multiple sites per brain to acquire data that reflects a fuller distribution of release levels in order to understand representative levels (e.g. Janezic et al 2013).

Moreover, control mice appear to vary significantly in the enhancement of DA release observed upon bath application of GABA receptor antagonists, with some control groups showing effects similar to *snca*^{-/-} mice (compare Fig. 1c and S1a to Fig. 8a) and others similar to *SNCA*⁺ mice (compare Fig. S1d and S1e to Fig. 8a), making it difficult to assess which model, if any, behaves as control, and in which light the effects of NPA and FC ought to be interpreted. How do the authors explain such variability between experimental groups, but not within groups? The authors should disclose more side-by-side comparisons of other metrics in *dat*-cre, *snca*^{-/-} and *SNCA*⁺ mice to determine how to best interpret the data.

RESPONSE: We remind the reviewer that our *SNCA+* and *Snca*^{-/-} mice are heterozygous for *Slc6a3*^{IRES-Cre}. In general, we observe ~20-30% increase in striatal DA release upon bath application of GABA receptor antagonists in WT mice, *Slc6a3*^{IRES-Cre} mice, and *Snca*^{-/-} mice, which is consistent with our previous findings in WT and *Slc6a3*^{IRES-Cre} mice (Lopes et al. 2019, Journal of Neuroscience 39, 1058-1065) and consistent with previous findings of others (Kramer et al. 2020, bioRxiv 941179; Brodник et al. 2019, ACS Chemical Neuroscience 10, 1978-1985). The effect size does inevitably vary between recording sites, subsets of data, and experimental set-ups. In the current manuscript, we have included two data sets where bath application of GABA receptor antagonists increases mean DA release in WT mice by ~40% (Fig. S1d and S1e). This is most likely because this subset of data was collected on a different experimental station to other data presented in this study. We can reassure that where data sets are directly compared in this study, they were done so using data acquired on a given set-up.

Please check the figure legend of Fig 1c and Fig 1f/g: it describes different Ns despite the mean traces looking identical. If controls are drawn from the exact same experimental group, the authors should state so clearly in the legend and refrain from using different representative DA transient traces. The same applies to Figs 2a,d,e,g,h, all of which include the same exact control data set, yet the representative traces change.

RESPONSE: We thank the reviewer for flagging this. We have now **revised Fig. 1 and Fig. 2 and the corresponding legend** to include the correct sample sizes used throughout, to indicate when controls are drawn from the exact same experimental groups, and to use identical representative DA transients where control data is repeated across experimental comparisons.

Note that the appropriate control groups for Fig. 2h should be slices maintained in DHBE+SKF, not slices maintained in DHBE only.

RESPONSE: We thank the reviewer for spotting this. We agree that the appropriate control group for this experimental data should be from slices maintained in DHβE and SKF89976A. We have now **included the appropriate control data** to resolve this (Fig. 2h).

Reviewers' Comments:

Reviewer #3:

Remarks to the Author:

I thank the authors for indulging my requests for clarifications, for addressing my concerns so thoroughly, and for editing their manuscript accordingly. I agree with the authors and other reviewers that their findings are novel and important, as they implicate astrocytes in the regulation of DA release via presynaptic GABA receptors and convincingly implicate GAT dysfunction in Parkinson's disease, and therefore support the publication of their manuscript in its current form without delay.